# BFS-Prover-V2: Scaling up Multi-Turn Off-Policy RL and Multi-Agent Tree Search for LLM Step-Provers

Ran Xin [* † 1]   Zeyu Zheng [* † 2]   Yanchen Nie [* † 3]   Kun Yuan [3]   Xia Xiao [1]

## Abstract

The integration of LLMs with automated theorem proving has shown immense promise, yet is constrained by challenges in scaling both training-time compute and inference-time compute. This paper introduces `BFS-Prover-V2`, a step-level theorem proving system designed to address this dual scaling problem. We present two primary innovations. The first is a multi-turn off-policy RL framework for continually improving the prover at training time. This framework, inspired by AlphaZero, utilizes a multi-stage expert iteration pipeline featuring adaptive tactic-level data filtering and periodic retraining to surmount the performance plateaus that typically curtail long-term RL in LLMs. The second innovation is a multi-agent system that scales reasoning capabilities at inference time, which employs a general reasoning model as a planner to iteratively decompose complex theorems into simpler sub-goals. This hierarchical approach substantially reduces the search space, enabling parallel provers to collaborate efficiently by leveraging a shared proof cache. We demonstrate that this dual approach yields SoTA results among tree-search provers on established formal math benchmarks, while remaining on par with the best whole-proof methods. `BFS-Prover-V2` achieves 95.08% and 41.4% on the miniF2F and ProofNet test sets respectively. While demonstrated in formal mathematics, the techniques presented in this work may be applied to other domains requiring long-horizon multi-turn reasoning and complex search. Our models and code have been open-sourced at https://github.com/ByteDance-Seed/BFS-Prover-V2.

---
[*]Equal contribution   [†]Work done at ByteDance Seed. [1]ByteDance Seed [2]Carnegie Mellon University [3]Peking University. Correspondence to: Xia Xiao <x.xiaxiao@bytedance.com>.

*Proceedings of the 43$^{rd}$ International Conference on Machine Learning*, Seoul, South Korea. PMLR 306, 2026. Copyright 2026 by the author(s).

## 1. Introduction

Automated Theorem Proving (ATP), a subfield of mathematical logic and automated reasoning, represents one of the foundational ambitions of computer science. The contemporary landscape of formal mathematics is increasingly dominated by interactive theorem provers (ITPs) or proof assistants. These systems, such as Coq, Isabelle, and Lean, require a human user to guide the proof process, but they automate significant deductive tasks and, most importantly, provide a machine-checkable guarantee of correctness (Geuvers, 2009). Among these, the Lean 4 programming language (Moura & Ullrich, 2021) has emerged as a particularly vibrant ecosystem. A key factor in its success is Mathlib (The mathlib Community, 2020), a vast, comprehensive, and community-driven library of formalized mathematics. Spanning over a million lines of code, Mathlib covers extensive areas of algebra, analysis, topology, and more, providing a rich foundation for both advanced mathematical research and the development of verified systems.

The rise of Lean 4 has coincided with the explosion in the capabilities of LLMs (OpenAI, 2023; Comanici et al., 2025; Seed et al., 2025), opening a new frontier in neuro-symbolic AI systems. The goal here is to integrate the intuitive yet powerful generation and search capabilities of LLMs with the absolute logical verification of formal systems. This research direction centers on a key feedback loop: an LLM proposes intuitive proof steps, the Lean compiler provides rigorous verification, and RL (Sutton, 2018) uses that verification to continuously improve the LLM's reasoning abilities (Yang et al., 2024b; Xin et al., 2024a; Polu et al., 2022; Han et al., 2021; Lample et al., 2022).

### 1.1. Motivation: A Duality of Scaling Challenges in Reasoning

The development of high-performance formal math provers, or any other reasoning agents, is contingent upon solving two fundamental and deeply interconnected scaling challenges.

**Training-time scaling.** This refers to the techniques required to continuously enhance a model's foundational capabilities and tactical intuitions via training. A common

and significant obstacle in applying RL to LLMs is the phenomenon of performance plateaus: after an initial phase of rapid improvement, models often stagnate, with their capabilities ceasing to grow despite continued training (Liu et al., 2025; Team et al., 2025; Yu et al., 2025; Yue et al., 2025; Guo et al., 2025; Seed et al., 2025; Xin et al., 2024a;b). Overcoming this limitation requires carefully designed algorithms that can sustain learning over extended periods, enabling the model to transition from mastering simple problems to tackling increasingly complex theorems.

**Inference-time scaling.** This addresses the method of using a trained model to solve previously unseen theorems. The primary bottleneck here is inefficient exploration in a vast search space. Specifically, pure tree search is often constrained by a lack of global planning ability and a search space that grows exponentially with proof depth. More generally, current inference paradigms suffer from the fact that search trajectories are independent, meaning that insights gained in one proof attempt are not shared with others. The challenge, therefore, is to design an inference architecture that incorporates planning and collaborative search to more effectively allocate computational resources (Baba et al., 2025; Zhou et al., 2025; Chen et al., 2025b; Jiang et al., 2022; Cao et al., 2025).

## 1.2. Our Contributions

This paper presents `BFS-Prover-V2`[1], a comprehensive training and inference system for neural theorem proving in Lean 4 that introduces novel solutions to the above scaling challenges. The primary contributions of this work are as follows:

**Novel RL Scaling Techniques at Training:** We develop a distillation-free multi-stage expert-iteration framework (Silver et al., 2018; Anthony et al., 2017), a form of off-policy RL, tailored for the domain of formal theorem proving. To sustain learning and overcome performance plateaus, we introduce a suite of specialized techniques within the RL pipeline. These include an adaptive, perplexity-based data filtering strategy at the tactic level, which creates an automated curriculum for the agent, and a periodic retraining mechanism that acts as a "soft reset" to escape local optima in the model parameter space and increase model scaling potential.

**Multi-Agent Tree Search System at Inference:** For inference-time scaling, we introduce a hierarchical reasoning architecture. A general-purpose reasoning model, termed the planner, provides global planning ability by iteratively decomposing complex theorems (or goals) into a sequence of more manageable subgoals. These subgoals

serve as tree search "checkpoints" with successful proof trajectories stored in a shared cache. This dramatically reduces search complexity by converting the total computational effort from a product of individual subgoal complexities to their sum.

**State-of-the-Art Empirical Results:** We validate the effectiveness and generalizability of our dual scaling approach on established benchmarks. In particular, `BFS-Prover-V2` achieves 95.08% on the miniF2F test set, largely surpassing previous LLM step-provers (Wu et al., 2024; Xin et al., 2025; Liang et al., 2025) and performing at state-of-the-art level compared with the best whole-proof generation models (Ren et al., 2025; Lin et al., 2025b; Wang et al., 2025). On ProofNet-test, it achieves 41.4%, reaching state-of-the-art level among step-level provers, and demonstrates robust generalization across distributions.

## 1.3. Conflict of Interest Disclosure

R.X., Z.Z., Y.N., and X.X. completed this work at ByteDance Seed. ByteDance Seed led the development of `BFS-Prover-V2`, the system evaluated in this paper.

## 2. The BFS-Prover-V2 System

This section details the two core components of `BFS-Prover-V2`: (i) a training pipeline, grounded in a Markov Decision Process (MDP) (Puterman, 1990) and scaled via adaptive filtering and periodic retraining; and (ii) an inference engine, which uses a planner-enhanced multi-agent search for hierarchical reasoning. These components build upon the foundation of `BFS-Prover-V1` (Xin et al., 2025) to specifically address the dual challenges of scaling at both training and inference time. We provide visual overviews of these components in Fig. 1 and 3, with practical implementation parameters and ablations detailed in Section 3.

### 2.1. A Step-Level Formulation: Theorem Proving as a Markov Decision Process

We formulate proof search in Lean 4 tactic mode as a Markov Decision Process (MDP), where an LLM-based prover (agent) interacts with the Lean proof checker (environment) to construct formal proofs. This formulation naturally captures the sequential, stateful nature of tactic-based formal theorem proving. Formally, we define the MDP tuple $\mathcal{M} = (S, A, P, R)$ as follows:

- **State Space** $S$**:** Each state $s \in S$ corresponds to a tactic state returned by the Lean compiler, comprising the current hypotheses (known facts) and target goals to be proven.

- **Action Space** $A$**:** An action $a \in A$ is a tactic string gen-

---

[1]Project Page: https://bfs-prover.github.io/V2/.

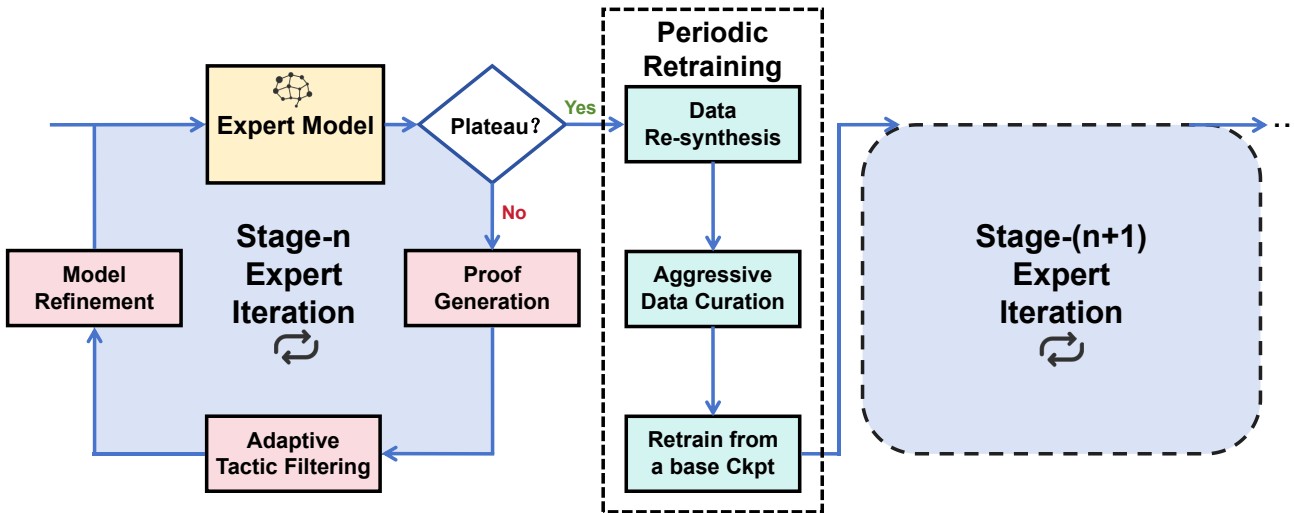

*Figure 1.* Overview of the training-time scaling up architecture. The process begins with a current expert model. The system then evaluates the model's performance to check for a plateau. If performance is improving, the model enters an inner **expert iteration loop**. Conversely, if performance has plateaued, the system triggers the outer **retraining loop**. The retraining loop yields an improved expert model serving as the starting point for the next cycle.

erated by the prover. Each tactic encodes a proof step, such as theorem application, term rewriting, or goal decomposition, that instructs the compiler to perform a specific deductive transformation.

- **Transition $P$:** The transition function $P(s' \mid s, a)$ is deterministically executed by the Lean proof checker. Given state $s$ and action $a$, the compiler either produces a successor state $s'$ if $a$ is applicable, or returns an error, resulting in a terminal failure state.

- **Reward Function $R$:** We employ sparse rewards where $R(s, a) = 1$ if and only if the state-action pair $(s, a)$ lies on a trajectory culminating in a successful proof. Otherwise, $R(s, a) = 0$.

This tactic-level stepwise interactive formulation fundamentally differs from whole-proof generation approaches (Ren et al., 2025; Lin et al., 2025b; Wang et al., 2025), which frame theorem proving as a one-shot code generation task from problem statements to complete proof scripts. While simpler, such approaches cannot adapt to intermediate proof states and lack integration with interactive theorem proving workflows (Welleck & Saha, 2023; Song et al., 2024). Our MDP-based approach, by design, trains an agent that functions as a genuine Lean copilot, capable of suggesting appropriate tactics at each step in the proof process (Yang et al., 2024c).

### 2.2. Scaling up Training: Multi-Stage Expert Iteration

The core training loop of `BFS-Prover-V2` is an expert iteration pipeline, which may be viewed as a variant of the

reinforcement learning algorithm used in AlphaZero (Anthony et al., 2017; Silver et al., 2018). Expert iteration was first introduced to formal theorem proving by GPT-f (Polu & Sutskever, 2020), and subsequent works including Hyper-Tree Proof Search (Lample et al., 2022) and AlphaProof (Hubert et al., 2025) have further developed AlphaZero-style approaches in this domain. This approach enables the system to learn and improve its theorem-proving capabilities from its own experience (Silver & Sutton, 2025). The process, illustrated in the inner loop of Fig. 1, includes two major alternating phases: proof generation and model refinement.

**Phase 1: Proof Generation:** The current best version of the LLM prover, referred to as the expert, is tasked with solving a large corpus of mathematical problems. The expert model is coupled with the best-first tree search (BFS) algorithm used in `BFS-Prover-V1` (Xin et al., 2025) to explore the vast space of possible proof trajectories. Each successful proof found during this phase constitutes a trajectory of (state, tactic) pairs. Across a single round of expert iteration, the system performs approximately $10^7$ tree searches, generating a massive synthetic dataset.

**Phase 2: Model Refinement:** The state-tactic pairs from the successful proof trajectories generated in the first phase are used to update the model's parameters. The updated model then becomes the new "expert" for the next round of iteration.

A central challenge in scaling the expert iteration pipeline or RL in general is managing the vast quantity and variable quality of the generated trajectories. Naively training on every successful trajectory quickly leads to diminishing returns, performance stagnation, and mode collapse (Liu

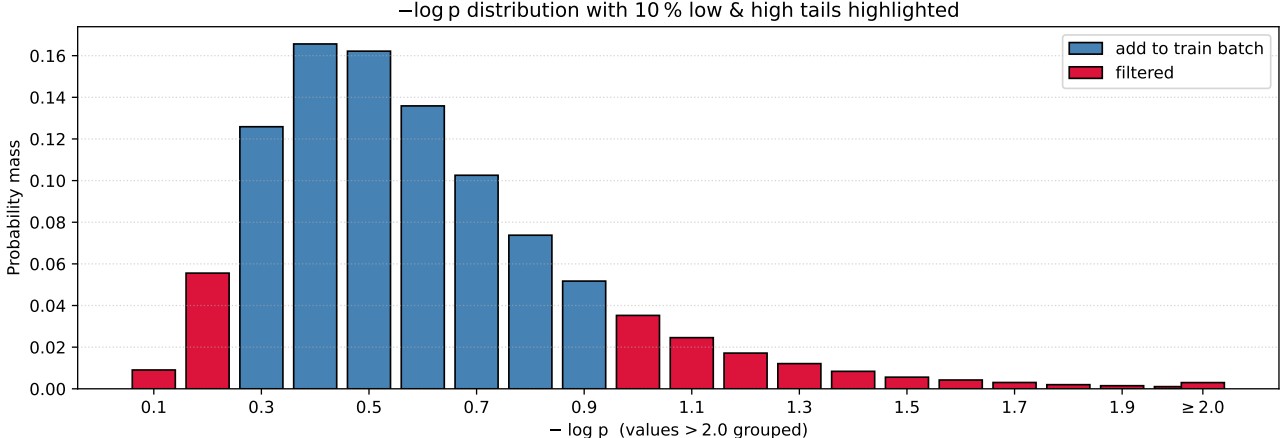

*Figure 2.* Tactic-Level Data Filtering Based on the Perplexity Distribution. This histogram shows the probability distribution of tactic perplexity (represented as normalized negative log-probability) from a single round of expert iteration. We filter out the low- and high-perplexity tails, shown in red.

et al., 2025; Sutton, 2018; Xin et al., 2025). To sustain improvement over many iterations, we introduce two key algorithmic innovations: a dynamic, fine-grained data filtering strategy and a periodic full-model retraining process. These techniques work in concert to form an automated curriculum that continuously improves the agent's capability over a long horizon. The overall architecture of this pipeline is illustrated in Fig. 1, and we detail each of these innovations in the following subsections.

### 2.2.1. ADAPTIVE TACTIC FILTERING: LEARNING FROM THE "JUST RIGHT" DATA

Instead of relying on coarse, problem-level filtering (Yu et al., 2025; Team et al., 2025), which often uses static metrics of difficulty, we adopt a more fine-grained approach at the tactic level. This strategy is guided by the empirical observation that the perplexity (negative log-probability) of tactics generated by the LLM roughly follows a right-skewed Gamma distribution. The distribution, shown in Fig. 2, can be divided into three distinct regions, each with different implications for learning:

- **The Low-Perplexity Tail:** This region corresponds to tactics for which the model has very high confidence. These are typically simple, "obvious" steps, such as basic simplification or applying a clear-cut hypothesis. Including these examples in the training batch offers no new learning signal; it merely reinforces what the model already knows well and can contribute to overfitting and a reduction in exploratory capacity.

- **The High-Perplexity Tail:** This region represents tactics that the model finds highly surprising. Our case studies reveal that these are often not instances of brilliant, non-obvious reasoning. Instead, they frequently

correspond to noisy or suboptimal choices, such as using a powerful, general-purpose tactic with many unnecessary parameters on a simple problem where a more direct tactic would suffice. These "fancy" operations can be detrimental to training, as they may teach the model to generate overly complex or irrelevant tactics, leading to hallucinations and degrading its core reasoning ability.

- **The Central Distribution:** This is the "goldilocks" zone. The tactics in this region are neither too easy nor too noisy. They represent steps that are challenging for the model but still within its grasp. By selectively training only on the data from this central part of the distribution, we ensure that the model is constantly learning at the edge of its capabilities.

**1. Low-Perplexity Tail**

*Trivial tactics* (**Discard**)

```
x:   ℝ
h₀:   x ∈ Set.Icc (π/2)(3π/2)
h₁:   0 ≤ cos x
⊢ π/2 ≤ x ∧ x ≤ 3π/2
```

```
exact h₀
```

**2. High-Perplexity Tail**

*Noisy/hallucinated tactics* (**Discard**)

```
p:   ℝ
h₀:   0 < p
hp:   ¬p = 0
h₂:   (800+5*p)*(7*p)=800*10*p
⊢ 7*p = 48*10
```

```
nlinarith [mul_pos h₀ (show 0 < p by linarith),
     mul_pos h₀ (by linarith: 0 < p/2), mul_pos h₀ (
     by linarith : 0 < p)]
```

### 3. Mid-Perplexity Zone

*Informative tactics* (**Keep**)

```
x:   ℝ
hx:   x = 250000
⊢ x = 2.5 * 10^5
```

```
norm_num [hx]
```

To further illustrate this filtering strategy, we present real-world examples of tactics falling into these three categories above. This adaptive filtering mechanism functions as a fully automated form of curriculum learning. It does not rely on any external (or predefined) metric of difficulty. Instead, it uses the model's own uncertainty (as measured by perplexity) as a dynamic signal of what constitutes valuable training data at its current stage of development. In our default inner-loop expert iterations we discard approximately the top and bottom 15% of the tactic perplexity distribution. This ensures a smooth and stable evolution of the model's internal policy distribution throughout training, enabling sustained growth in performance.

### 2.2.2. PERIODIC RETRAINING: A "SOFT RESET" TO ESCAPE LOCAL OPTIMA

Even with adaptive filtering, performance eventually plateaus as the prover's tactic preferences become increasingly rigid, causing it to overfit to a narrow set of proof patterns and settle into a local optimum. It becomes very good at solving problems in particular ways, but loses the ability to discover novel approaches required for harder or new problems. To escape local optima and reinvigorate the learning process, we introduce a periodic "soft reset" procedure. This constitutes a multi-stage expert-iteration process designed to increase the model's entropy and reset its exploratory potential without losing the competence it has already gained. The procedure is as follows:

1. **Re-synthesis and De-noise:** We re-run the current expert prover on the full historical problem set to re-generate all proofs using its improved policy. Since the prover at this stage is substantially stronger than the checkpoints that produced the earlier trajectories, the newly synthesized proofs are typically shorter, structurally cleaner, and contain fewer spurious steps. This pass serves as a model-driven denoising phase: it replaces outdated trajectories with higher-quality ones and removes redundant or circuitous reasoning patterns that accumulated during earlier, less capable iterations.

2. **Aggressive Data Curation:** The new, higher-quality proofs generated in the data re-synthesis phase are then subjected to an aggressive version of the tactic-level perplexity filtering described above. Specifically, we discard approximately the top and bottom 20% of the tactic perplexity distribution, compared with the 15% used in inner-loop iterations, retaining only the most

crucial and informative tactic steps.

3. **Retrain from a base Checkpoint:** The existing training data is completely replaced by this new, highly curated, and compact dataset. A fresh model instance is then initialized from the base checkpoint and trained from scratch on this refined data.

The resulting model initially exhibits a temporary drop in performance. This is expected, as it has been trained on a smaller, more focused dataset and has forgotten some of its previous stylistic biases. However, this new model possesses a significantly higher exploratory potential. When it is reintroduced into the expert iteration loop, its increased capacity for exploration allows it to discover novel proof strategies that were inaccessible to the previous, over-specialized model. Consequently, its performance rapidly recovers and then surpasses the previous peak.

### 2.3. Scaling up Inference: Planner-Enhanced Multi-Agent Search

Many complex proofs in mathematics are not found through a linear sequence of simple deductions but rather through a hierarchical process of identifying and proving crucial intermediate results. Inspired by Draft, Sketch, and Prove (Jiang et al., 2022; Cao et al., 2025) we introduce a hierarchical inference architecture, shown in Fig. 3, that divides the labor of theorem proving between two distinct agents: a high-level planner and a low-level prover.

**Planner:** This is a general-purpose reasoning LLM tasked with goal decomposition. Given the current theorem statement and proof progress, its role is not to generate a specific tactic but to propose a high-level plan that includes a series of intermediate subgoals. By formulating these subgoals, the planner effectively transforms a single, monolithic, and potentially intractable search problem into a structured sequence of smaller, more manageable ones. This decomposition substantially reduces the dimensionality of the search space that the prover must explore.

**Prover:** This is the specialized LLM tactic generator trained via our multi-stage expert iteration pipeline described in Section 2.2. It receives one subgoal at a time from the planner and uses its learned policy, in conjunction with Best-first tree search algorithm (Xin et al., 2025), to find a formal proof for that specific subgoal.

This division of labor mirrors the cognitive workflow of a human mathematician, who might first sketch out the high-level structure of a proof by identifying key lemmas (the planner's role) and then proceed to work out the detailed, step-by-step deductions for each lemma (the prover's role). This hierarchical structure is a powerful architectural pattern for tackling complex reasoning tasks.

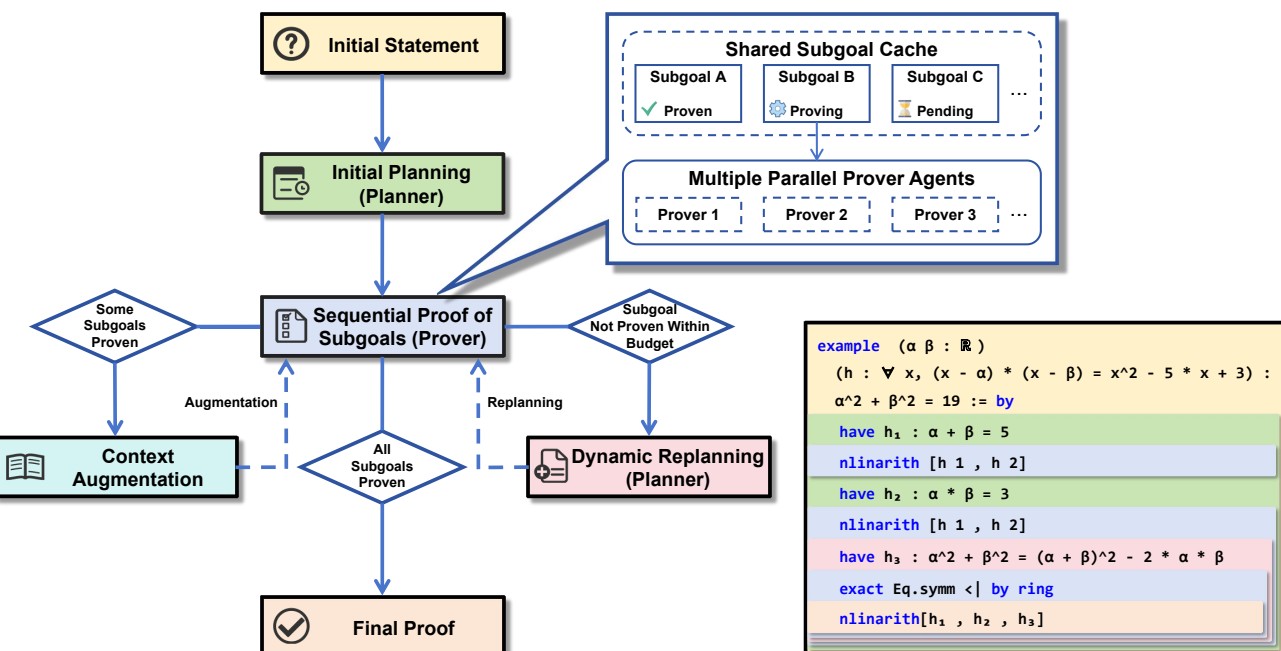

*Figure 3.* Overview of the multi-agent tree search architecture. The **Planner** agent decomposes the main theorem into a sequence of simpler subgoals, which are managed in a **Shared Subgoal Cache** and solved in parallel by multiple **Prover** agents. Successfully proven subgoals augment the main proof's context, while failures trigger a **Dynamic Replanning** loop. The inset provides an example, demonstrating how proving intermediate lemmas ($h_1$, $h_2$, $h_3$) facilitates the proof of the final goal.

### 2.3.1. OPERATIONAL MECHANICS OF PLANNER-GUIDED SEARCH

As shown in Fig. 3, the interaction between the planner and the prover system unfolds in a dynamic loop, allowing the plan to be revised as proof search progresses:

1. **Initial Planning:** At the start of a proof attempt, the planner is queried with the main theorem statement. It returns a list of proposed subgoals, formatted as Lean `have` statements.

2. **Sequential Proof of Subgoals:** The prover system addresses the subgoals one by one. It takes the first subgoal in the queue and initiates tree search to find its proof.

3. **Context Augmentation:** When a subgoal is solved, its statement is incorporated into the proof context and becomes available as a hypothesis for all remaining subgoals and the main theorem itself.

4. **Dynamic Replanning:** If the prover exhausts its search budget on a subgoal, which may occur either because the subgoal is intrinsically difficult or because the planner previously produced an incorrect subgoal, the system does not terminate. Instead, the planner is invoked again with an augmented input containing the theorem statement and all previously proven subgoals. The planner then generates a revised decomposition

that typically corrects, refines, or further subdivides the subgoal where search stalled.

This dynamic and iterative loop between planning and proving makes the `BFS-Prover-V2` system resilient to getting stuck, effectively scaling its inference-time capabilities to tackle complex theorems that would be intractable for a monolithic tree search.

### 2.3.2. MULTI-AGENT COLLABORATION VIA FOCUSED PARALLELISM AND SHARED SUBGOAL CACHE

To further scale inference-time compute, we deploy the planner–prover architecture in a multi-agent setting in which several prover instances run in parallel. These agents jointly execute the subgoal sequence proposed by the planner, coordinated through two design principles: focused parallelism and a shared subgoal cache.

**Focused parallelism:** Rather than distributing different subgoals in parallel across agents, all prover instances allocate their full search budget to a single active subgoal before the system advances to the next. This sequential execution concentrates search effort on difficult reasoning bottlenecks where only a subset of provers would need substantially more time to progress and avoids wasted compute on downstream subgoals that would be invalidated if an earlier step fails and triggers a replan.

**Shared Subgoal Cache:** This cache is the central commu-

nication and state-tracking mechanism, shared across all parallel prover instances. It stores the full sequence of sub-goals generated by the planner, tracks the real-time status of each subgoal (e.g., pending, proving, proven), and records the proof for any solved subgoal.

This architecture creates a cooperative sprint for each lemma in the plan. When a new subgoal is designated as the active target, all prover agents begin independent tree searches for that single subgoal in parallel. As soon as the first agent finds a valid proof, it writes the result to the shared cache. The subgoal cache signals all other agents to terminate their search, preventing redundant computation. The entire group of agents then proceeds to the next subgoal in the sequence.

## 3. Experiments

### 3.1. Practical Implementation

**Base model and Data:** Our prover is built upon Qwen2.5-Math-7B and Qwen2.5-32B (Yang et al., 2024a), which serve as the base for our policy optimization. The multi-stage expert iteration process was initialized from the checkpoint `BFS-Prover-V1` (Xin et al., 2025). To construct a large-scale training corpus, we autoformalized the NuminaMath-CoT and NuminaMath-1.5 datasets (Li et al., 2024) by prompting GPT-4o and Claude 3.7 Sonnet, augmented with Lean 4 compiler feedback. Together with Goedel-Pset-V1 (Lin et al., 2025a), this yields roughly 3 million formal statements. Prompts used for autoformalization can be found in Section C.1. All experiments are conducted in Lean v4.10.0 with LeanDojo (Yang et al., 2024c).

**Training setup:** After each expert iteration round, we refine the policy LLM using one of two supervised fine-tuning (SFT) strategies, selected based on the outcome of the round. In the inner expert-iteration loop, we fine-tune the current best checkpoint for one epoch with cosine learning rate decay from $5 \times 10^{-6}$ to $1 \times 10^{-7}$. When periodic retraining is triggered (Section 2.2), we train for three epochs with a larger learning rate that decays from $2 \times 10^{-5}$ to $1 \times 10^{-6}$. Both strategies use a global batch size of 1024.

**Inference configuration:** Our inference pipeline combines a low-level prover with a high-level planner, as detailed in Section 2.3. Prover agents perform best-first search (BFS) following the `BFS-Prover-V1` implementation (Xin et al., 2025). Unless stated otherwise, we use a sampling temperature of 1.3, an expansion width of 3, and a length-normalization factor of 2.0 during expert iterations. For the planner, we use Gemini 2.5 Pro; other general-purpose reasoning models can reach similar performance given suitable prompts. Planner prompts are provided in Section C.2.

*Table 1.* Comparison with other leading theorem provers. For tree-search provers, the Budget column denotes search passes $\times$ expansion width or parallel setting $\times$ per-instance timeout in seconds; for whole-proof provers, it denotes the reported sample budget or pass@K. [†] denotes concurrent work.

| Prover Method | Budget | miniF2F | ProofNet |
|---|---|---|---|
| *Tree-search provers* | | | |
| InternLM2.5-Step-7B | $256 \times 32 \times 600$ | 65.9% | $\approx 27\%$ |
| Hunyuan-Prover-7B | $600 \times 8 \times 400$ | 68.4% | - |
| BFS-Prover-V1-7B | $2048 \times 2 \times 600$ | 70.8% | - |
| | accumulative | 73.0% | - |
| MPS-Prover-7B[†] | $64 \times 4 \times 800 \times 8$ | 72.5% | - |
| | accumulative | 75.8% | - |
| **BFS-Prover-V2-7B** | accumulative | 82.4% | 34.4% |
| w/ Planner | accumulative | 92.6% | - |
| **BFS-Prover-V2-32B** | $8192 \times 3 \times 600$ | 85.3% | - |
| | $8192 \times 8 \times 600$ | - | 40.3% |
| | accumulative | 86.1% | 41.4% |
| w/ Planner | accumulative | 95.1% | - |
| *Whole-proof provers* | | | |
| DeepSeek-Prover-V2-7B | 8192 / 1024 | 82.0% | 29.6% |
| w/ Prover Agent | 260 | 82.8% | - |
| DeepSeek-Prover-V2-671B | 8192 / 1024 | 88.9% | 37.1% |
| Kimina-Prover-72B[†] | 1024 | 87.7% | - |
| w/ TTRL search | accumulative | 92.2% | - |
| Goedel-Prover-V2-7B[†] | 8192 | 90.2% | - |
| w/ Prover Agent | 260 | 86.5% | - |
| Goedel-Prover-V2-32B[†] | 8192 | 92.2% | - |
| w/ Self-correction | 1024 | 92.6% | - |
| Delta-Prover[†] | accumulative | 95.9% | - |
| Seed-Prover[†] | accumulative | **99.6%** | - |

## 3.2. Benchmark Results

We evaluated `BFS-Prover-V2` on two primary benchmarks: miniF2F (Zheng et al., 2021), which targets high-school mathematical Olympiad problems (in-distribution), and ProofNet (Azerbayev et al., 2023), which covers undergraduate textbook level mathematics and serves as a rigorous test for out-of-distribution (OOD) generalization.

As detailed in Table 1, our system establishes a new state of the art among step-level tree-search provers. On the miniF2F-test set, our 32B model with the planner achieves an accuracy of 95.08% (95.49% on miniF2F-valid), while the 7B model reaches 92.6%.

Crucially, our approach demonstrates robust OOD generalization on ProofNet-test. The 32B model achieves 41.4%, and the 7B model reaches 34.4% via pure tree search, notably surpassing the 671B DeepSeek-Prover-V2 (37.1%) despite being 20 times smaller, and its 7B variant (29.6%), respectively. We attribute this superior OOD performance to the inherent flexibility of step-level proving: unlike whole-proof generation models that rely heavily on the training distribution, a trained step-prover can adapt its exploration strategy by adjusting search parameters at test time to match problem distributions, enabling effective transfer without retraining.

## 3.3. Further Analysis on Training

We report training-time ablations that justify the utility of tactic-level data curation, the critical role of periodic retraining in escaping local optima, and the motivation for scaling to larger base models.

**Perplexity-based tactic filtering:** We investigated the impact of our filtering strategy during the multi-stage expert iterations. We conducted an ablation on Checkpoint 3 in Fig. 4 (derived after the first retraining phase). The training corpus consisted of 459,540 human state-tactic pairs extracted from GitHub Lean4 repositories. Without tactic filtering, the combination with synthetic expert iteration data yielded 857,897 state-tactic pairs. Training on this unfiltered set resulted in a validation loss of 0.5597 and a miniF2F test score of 75%. In contrast, applying our perplexity-based filtering reduced the dataset to 660,254 high-quality samples. Despite the smaller data volume, this filtered run (Checkpoint 4) resulted in a higher validation loss (0.6211) but a superior test score of 76.63%. Further validation on checkpoint 4 confirmed the approach's effectiveness: training with all expert iteration data resulted in performance degradation to 75.81%, while continued filtering improved performance to 77.04%.

**Periodic retraining:** Performance plateaus appeared at several checkpoints during training, where continued expert iteration did not improve and sometimes even reduced performance. Our periodic retraining mechanism consistently overcame these local optima. At checkpoint 2, performance stagnated at 75.41 percent. After retraining, accuracy briefly decreased to 75.00% but then increased to 76.64% in the next iteration. A similar pattern occurred at checkpoint 6: accuracy was 78.28% before retraining, declined to 77.05% immediately after retraining, and then reached 79.92% after two additional iterations. Crucially, this phenomenon is scalable: the larger 32B model exhibited a similar plateau at Checkpoint 19 (85.25%), dropped to 84.02% after retraining, and later reached 86.07%. Fig. 4 presents the corresponding progression over time. In practice, we trigger a soft reset when the expert-iteration loop produces very few newly solved problems and downstream miniF2F accuracy stops improving or begins to decline. The reset then re-synthesizes the proof corpus with the current expert, applies the more aggressive filtering described in Section 2.2, and retrains from the base checkpoint to restore exploration while preserving the accumulated high-quality proof data.

**Model scaling:** We observed diminishing returns for the 7B prover as its improvement rate on the training corpus and miniF2F began to saturate, suggesting a capacity limitation inherent to the model size. To verify the scalability of our training recipe, we extended the multi-stage expert iteration experiments to the Qwen2.5-32B model. Scaling to 32B parameters yielded immediate gains even without additional data: checkpoints 16 and 17, trained on identical corpus, achieved 82.38% and 82.79% respectively. Critically, the 32B model demonstrated superior out-of-distribution generalization on ProofNet (41.4% vs. 34.4% for 7B), indicating that increased model capacity enhances transfer to novel mathematical domains beyond the training distribution.

## 3.4. Further Analysis on Inference

**Computational budget:** We report *accumulative* performance as the union of problems solved across a small grid of search hyperparameters, with pass@8192 evaluated separately for each configuration. The grid covers branching factors $\{2, 3, 4, 8\}$ and depth rewards $\{0, 1, 2\}$. To show that performance is not solely due to accumulation, Table 1 also reports single-configuration results for BFS-Prover-V2-32B. When the planner is enabled, pass@$K$ refers to $K$ cumulative prover instances sharing the per-problem sub-goal cache rather than $K$ independent searches; without the planner, pass@$K$ corresponds to $K$ independent tree searches. Each prover instance is run with a 600s wall-time budget that includes all Lean interactions.

**Planner effectiveness:** The integration of the Planner agent provides a massive performance boost at both model scales. The 7B model's performance jumps from 82.4% to 92.6% with the planner, while the 32B model improves from 86.1% to 95.1%. Notably, the planner narrows the performance gap

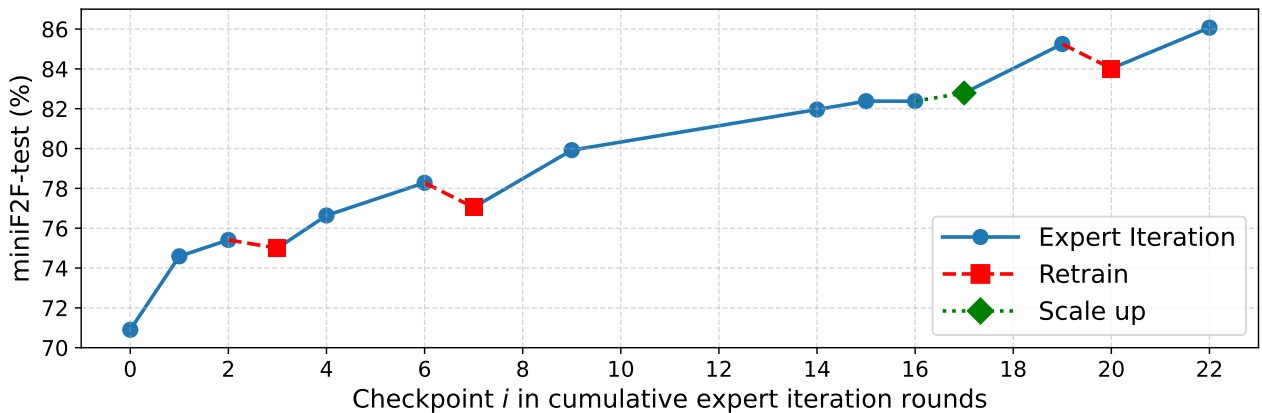

*Figure 4.* Sustained Performance Improvement through Expert Iteration and Periodic Retraining. This graph plots the prover's performance on the miniF2F benchmark vs. the expert iteration rounds.

between model scales, demonstrating that effective search strategies can partially compensate for raw model capacity.

**Subgoal cache:** The shared subgoal cache is critical for reducing computational complexity. It effectively transforms the search complexity from the product of individual subgoal search spaces to their sum. Empirically, for `amc12a_2008_p25` and `mathd_algebra_17` in miniF2F-test, a planner-only setup without caching failed to find a solution after 8,192 cumulative prover instances. In contrast, with the shared subgoal cache, the system consistently solved these problems using fewer than 512 cumulative instances, preventing redundant computation on established subgoals.

## 4. Conclusion

The primary contributions of this work are the design, implementation, and empirical validation of a holistic system for scaling LLM-based step-provers. On the training side, our multi-stage expert iteration pipeline overcomes common performance plateaus and enables sustained improvement over an extended training period. On the inference side, by leveraging a planner agent for subgoal decomposition and a shared subgoal cache for collaborative search, our system transforms intractable search spaces into manageable sequences of tasks. Empirically, `BFS-Prover-V2` not only achieves state-of-the-art-level performance on miniF2F but also exhibits robust OOD generalization on ProofNet, providing strong evidence for the efficacy of our approach.

## 5. Limitations

The results in this paper should be interpreted with several constraints in mind. First, the training pipeline is compute-intensive. Full-parameter SFT requires approximately 250–

300 A100 GPU hours per epoch for the 7B prover and 1,500–1,800 A100 GPU hours per epoch for the 32B prover; a standard expert-iteration update uses roughly one epoch, while retraining after a soft reset uses three epochs. This cost limited our ability to repeat the full pipeline across random seeds or to run every ablation as an independent end-to-end training run. We therefore analyze the long-horizon dynamics mainly through the observed training trajectory and the plateau-recovery events in Fig. 4.

Second, the current training utilizes only successful proof trajectories after tactic-level filtering. Failed searches, compiler errors, and hard negative tactics likely contain useful information about the boundary of the policy, but they are not yet incorporated systematically. Improving the use of these negative signals is a natural direction for making expert iteration more data-efficient.

Finally, the planner and prover are optimized separately. This modular design makes the system easy to inspect and allows us to use strong general-purpose reasoning models as planners, but it also means that decomposition decisions are not co-trained with the proof policy or the subgoal cache. In addition, the current multi-agent search shares solved subgoals across prover instances but does not parallelize the expansion of a single tree search. A more integrated system could adapt plans to the prover's failure modes and exploit finer-grained intra-search parallelism.

## Acknowledgements

We would like to thank Ming Ding from ByteDance Seed for his insightful discussions throughout this project.

Kun Yuan is supported by the National Key Research and Development Program of China (No. 2024YFA1012902).

## Impact Statement

This paper presents work whose goal is to advance the field of Machine Learning. There are many potential societal consequences of our work, none of which we feel must be specifically highlighted here.

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

# A. Case Studies

## A.1. Proof Conciseness and Tactic Proficiency

A primary advantage of our step-level proof approach over the whole-proof paradigm is a dramatic reduction in proof length, which arises from the interactive nature of our method. By engaging with the Lean environment step by step, our model captures and leverages fine-grained tactic state information. This iterative feedback loop significantly improves its ability to employ powerful, high-level tactics such as `simp`, `linarith`, and `nlinarith`, enabling it to solve complex goals in a single step rather than through lengthy, explicit derivations.

To illustrate this contrast, we present a comparative analysis of proofs for two problems, `aime_1984_p7` and `amc12_2000_p1`, generated by `BFS-Prover-V2`, `DeepSeek-Prover-V2`, and `Goedel-Prover-V2`.

---

### aime_1984_p7

For the problem `aime_1984_p7`, `BFS-Prover-V2` produces a remarkably concise proof, where a single tactic `simp [h_1, h_0]` effectively replaces over 2000 lines that appear in numerous `have` blocks within `DeepSeek-Prover-V2`'s proof. `Goedel-Prover-V2` likewise relies on a long chain of explicit `have` blocks that unfold the recurrence step by step.

**BFS-Prover-V2 (step-level proof):**

```
theorem aime_1984_p7
  (f : ℤ → ℤ)
  (h₀ : ∀ n, 1000 ≤ n → f n = n - 3)
  (h₁ : ∀ n, n < 1000 → f n = f (f (n + 5))) :
  f 84 = 997 := by
  simp [h₁, h₀]
```

**DeepSeek-Prover-V2 (whole-proof):**

```
theorem aime_1984_p7 (f : ℤ → ℤ) (h₀ : ∀ n, 1000 ≤ n → f n = n - 3)
    (h₁ : ∀ n, n < 1000 → f n = f (f (n + 5))) : f 84 = 997 := by
  have h₂ : f 1004 = 1001 := by
    have h₂₁ : f 1004 = 1004 - 3 := by
      apply h₀
      <;> norm_num
    rw [h₂₁]
    <;> norm_num

  have h₃ : f 1003 = 1000 := by
    have h₃₁ : f 1003 = 1003 - 3 := by
      apply h₀
      <;> norm_num
    rw [h₃₁]
    <;> norm_num

  have h₄ : f 1002 = 999 := by
    have h₄₁ : f 1002 = 1002 - 3 := by
      apply h₀
      <;> norm_num
    rw [h₄₁]
    <;> norm_num

  -- ... (approximately 2000 lines of similar calculations omitted) ...

  have h₁₉₃ : f 89 = 998 := by
    have h₁₉₃₁ : f 89 = f (f (89 + 5)) := by
```

```
      apply h₁
      <;> norm_num
    rw [h₁₉₃₁]
    have h₁₉₃₂ : f (89 + 5) = f 94 := by norm_num
    rw [h₁₉₃₂]
    rw [h₁₉₂]
    <;> norm_num
    <;> linarith

  have h₁₉₄ : f 84 = 997 := by
    have h₁₉₄₁ : f 84 = f (f (84 + 5)) := by
      apply h₁
      <;> norm_num
    rw [h₁₉₄₁]
    have h₁₉₄₂ : f (84 + 5) = f 89 := by norm_num
    rw [h₁₉₄₂]
    rw [h₁₉₃]
    <;> norm_num
    <;> linarith

  rw [h₁₉₄]
  <;> norm_num
```

### Goedel-Prover-V2 (whole-proof):

```
theorem aime_1984_p7
  (f : ℤ → ℤ)
  (h₀ : ∀ n, 1000 ≤ n → f n = n - 3)
  (h₁ : ∀ n, n < 1000 → f n = f (f (n + 5))) :
  f 84 = 997 := by
  have h2 : f 999 = 998 := by
    have h2₁ : f 999 = f (f (999 + 5)) := by
      apply h₁
      <;> norm_num
    have h2₂ : f (999 + 5) = (999 + 5 : ℤ) - 3 := by
      apply h₀
      <;> norm_num
    have h2₃ : f (999 + 5) = 1001 := by
      rw [h2₂]
      <;> norm_num
    have h2₄ : f 999 = f 1001 := by
      rw [h2₁, h2₃]
      <;> norm_num
    have h2₅ : f 1001 = (1001 : ℤ) - 3 := by
      apply h₀
      <;> norm_num
    have h2₆ : f 1001 = 998 := by
      rw [h2₅]
      <;> norm_num
    rw [h2₄, h2₆]
    <;> norm_num

  -- ... (approximately 140 lines establishing f 998, f 997, and the parity recurrence
     f (999 - 5k) by induction omitted) ...

  have h6 : f 84 = 997 := by
    have h₆₁ : f (999 - 5 * (183 : ℤ)) = if (183 : ℕ) % 2 = 0 then 998 else 997 := h5
    183
    have h₆₂ : (999 - 5 * (183 : ℤ) : ℤ) = 84 := by norm_num
    rw [h₆₂] at h₆₁
    have h₆₃ : (183 : ℕ) % 2 = 1 := by norm_num
```

```
    rw [if_neg] at h₆₁ <;> norm_num [h₆₃] at h₆₁ ⊢
  exact h6
```

## amc12_2000_p1

For the problem `amc12_2000_p1`, `BFS-Prover-V2` finishes the proof with a concise application of `nlinarith`, replacing the roughly 80 lines of explicit case analysis over the unknowns *i*, *m*, and *o* required in `DeepSeek-Prover-V2`'s proof. `Goedel-Prover-V2` follows the same exhaustive strategy, enumerating the divisors of 2001 for each unknown.

### BFS-Prover-V2 (step-level proof):

```
theorem amc12_2000_p1
  (i m o : ℕ)
  (h₀ : i ≠ m ∧ m ≠ o ∧ o ≠ i)
  (h₁ : i*m*o = 2001) :
  i+m+o ≤ 671 := by
  obtain ⟨hi, hm, h₂⟩:= h₀
  obtain _ | _ | _ | i := i <;> obtain _ | _ | _ | m := m <;> obtain _ | _ | _ | o :=
    o <;>try omega
  any_goals
    ring_nf at h₁ ⊢; nlinarith;
  ring_nf at * <;> try omega
  nlinarith [mul_nonneg (by positivity : 0 ≤ i) (by positivity : 0 ≤ o), mul_nonneg (
    by positivity : 0 ≤ m) (by positivity : 0 ≤ i), mul_nonneg (by positivity : 0 ≤ o)
      (by positivity : 0 ≤ m)]
```

### DeepSeek-Prover-V2 (whole-proof):

```
theorem amc12_2000_p1 (i m o : ℕ) (h₀ : i ≠ m ∧ m ≠ o ∧ o ≠ i) (h₁ : i * m * o = 2001)
    :
    i + m + o ≤ 671 := by

    -- ... (approximately 80 lines of properties related to unknowns i, m, and o
    omitted) ...

  have h_main : i + m + o ≤ 671 := by
    have h₈ : i = 1 ∨ i = 3 ∨ i = 23 ∨ i = 29 ∨ i = 69 ∨ i = 87 ∨ i = 667 ∨ i = 2001
    := by
      have h₈₁ : i | 2001 := h₂
      have h₈₂ : i ≤ 2001 := h₅
      have h₈₃ : i > 0 := h_pos_i
      have h₈₄ : i = 1 ∨ i = 3 ∨ i = 23 ∨ i = 29 ∨ i = 69 ∨ i = 87 ∨ i = 667 ∨ i =
    2001 := by
        have h₈₅ : i | 2001 := h₈₁
        have h₈₆ : i ≤ 2001 := h₈₂
        interval_cases i <;> norm_num at h₈₅ ⊢ <;> omega
      exact h₈₄
    have h₉ : m = 1 ∨ m = 3 ∨ m = 23 ∨ m = 29 ∨ m = 69 ∨ m = 87 ∨ m = 667 ∨ m = 2001
    := by
      have h₉₁ : m | 2001 := h₃
      have h₉₂ : m ≤ 2001 := h₆
      have h₉₃ : m > 0 := h_pos_m
      have h₉₄ : m = 1 ∨ m = 3 ∨ m = 23 ∨ m = 29 ∨ m = 69 ∨ m = 87 ∨ m = 667 ∨ m =
    2001 := by
        have h₉₅ : m | 2001 := h₉₁
```

```
      have h₉₆ : m ≤ 2001 := h₉₂
      interval_cases m <;> norm_num at h₉₅ ⊢ <;> omega
    exact h₉₄
  have h₁₀ : o = 1 ∨ o = 3 ∨ o = 23 ∨ o = 29 ∨ o = 69 ∨ o = 87 ∨ o = 667 ∨ o = 2001
  := by
    have h₁₀₁ : o | 2001 := h₄
    have h₁₀₂ : o ≤ 2001 := h₇
    have h₁₀₃ : o > 0 := h_pos_o
    have h₁₀₄ : o = 1 ∨ o = 3 ∨ o = 23 ∨ o = 29 ∨ o = 69 ∨ o = 87 ∨ o = 667 ∨ o =
  2001 := by
      have h₁₀₅ : o | 2001 := h₁₀₁
      have h₁₀₆ : o ≤ 2001 := h₁₀₂
      interval_cases o <;> norm_num at h₁₀₅ ⊢ <;> omega
    exact h₁₀₄
  rcases h₈ with (rfl | rfl | rfl | rfl | rfl | rfl | rfl | rfl) <;>
  rcases h₉ with (rfl | rfl | rfl | rfl | rfl | rfl | rfl | rfl) <;>
  rcases h₁₀ with (rfl | rfl | rfl | rfl | rfl | rfl | rfl | rfl) <;>
  norm_num [mul_assoc, mul_comm, mul_left_comm] at h₁ h₀ ⊢ <;>
  (try omega) <;>
  (try
    {
      norm_num at h₀ ⊢ <;>
      (try omega) <;>
      (try
        {
          ring_nf at h₁ ⊢ <;>
          omega
        })
    }) <;>
  (try
    {
      norm_num at h₀ ⊢ <;>
      (try omega) <;>
      (try
        {
          ring_nf at h₁ ⊢ <;>
          omega
        })
    }) <;>
  (try
    {
      norm_num at h₀ ⊢ <;>
      (try omega) <;>
      (try
        {
          ring_nf at h₁ ⊢ <;>
          omega
        })
    }) <;>
  (try
    {
      norm_num at h₀ ⊢ <;>
      (try omega) <;>
      (try
        {
          ring_nf at h₁ ⊢ <;>
          omega
        })
    })
  <;>
  (try omega)
  <;>
  (try
```

```
    {
      ring_nf at h₁ ⊢ <;>
      omega
    })

  exact h_main
```

**Goedel-Prover-V2 (whole-proof):**

```
theorem amc12_2000_p1
  (i m o : ℕ)
  (h₀ : i ≠ m ∧ m ≠ o ∧ o ≠ i)
  (h₁ : i*m*o = 2001) :
  i+m+o ≤ 671 := by
  have h_main : i + m + o ≤ 671 := by
    have h₂ : i | 2001 := by
      use m * o
      linarith
    have h₃ : m | 2001 := by
      use i * o
      linarith
    have h₄ : o | 2001 := by
      use i * m
      linarith
    -- ... (approximately 120 lines enumerating the divisors of 2001 for i, m, and o
    and discharging every case omitted) ...
    rcases h₈ with (rfl | rfl | rfl | rfl | rfl | rfl | rfl | rfl) <;>
      rcases h₉ with (rfl | rfl | rfl | rfl | rfl | rfl | rfl | rfl) <;>
        rcases h₁₀ with (rfl | rfl | rfl | rfl | rfl | rfl | rfl | rfl) <;>
          (try norm_num at h₁ ⊢ <;> try omega) <;>
          (try simp_all [mul_assoc] <;> ring_nf at * <;> norm_num at * <;> omega)
  exact h_main
```

## A.2. Novel Proof Strategies

Another significant advantage of our step-level proof approach is its ability to discover novel proof strategies that whole-proof or human-proof methods typically would not consider. By exploring the proof space progressively, our system can identify non-obvious connections and construct solutions that are both elegant and insightful.

We illustrate this capability by examining the problems imo_1963_p5 and algebra_amgm_sum1toneqn_prod1tonleq1, each of which highlights a distinct advantage of our approach.

---

**imo_1963_p5 - Part 1**

For the problem imo_1963_p5, our model provides a step-level proof, DeepSeek-Prover-V2 and Goedel-Prover-V2 provide generated whole-proof variants, and the Compfiles dataset provides a human-proof version. Notably, both whole-proof and human-proof approaches employ similar strategies: multiplying both sides of the equation by $2 \cdot \sin(\pi/7)$, then applying sum-to-product trigonometric identities for simplification. In contrast, BFS-Prover-V2 develops an entirely different approach: first transforming the left side of the equation into a polynomial in $\cos(\pi/7)$ using double and triple angle formulas, then proving that $\cos(\pi/7)$ satisfies the corresponding polynomial equation.

**BFS-Prover-V2 (step-level proof):**

```
theorem imo_1963_p5 :
  Real.cos (π / 7) - Real.cos (2 * π / 7) + Real.cos (3 * π / 7) = 1 / 2 := by
```

---

```
have x : Real.pi / 7 = Real.pi / 7 * 1 := by ring
have h : 3 * Real.pi / 7 = Real.pi - 4 * Real.pi / 7 := by ring
rw [h, cos_sub] <;> norm_num
have h2 := cos_two_mul (Real.pi / 7)
have h3 := cos_three_mul (π / 7)
rw [show 4 * Real.pi / 7 = Real.pi - 3 * Real.pi / 7 by ring,
  cos_sub]
simp [h2, h3, cos_two_mul, sin_pi, cos_pi]
ring_nf at h2 h3 ⊢
norm_num [h2, h3, cos_pi_div_two]
ring_nf
  <;> have h4 := cos_pi
  <;> simp [h4]
ring_nf at * <;> norm_num
rw [← sub_eq_zero]
nth_rewrite 1 [← sub_eq_zero]
ring_nf
apply eq_of_sub_eq_zero
let y := cos (Real.pi * (1 / 7))
have:= cos_three_mul (Real.pi * (1 / 7))
ring_nf at *
apply eq_of_sub_eq_zero
clear this h3 h2
apply eq_of_sub_eq_zero
have := cos_three_mul (Real.pi * (1 / 7))
field_simp [mul_assoc] at *
on_goal 1 => ring
replace : Real.pi * (1 / 7 : ℝ) = Real.pi / 7 := by ring
try rw [this]; norm_num
have h5 := cos_three_mul (Real.pi / 7)
have : 3 * (Real.pi / 7) = Real.pi - 4 * (Real.pi / 7) := by ring
simp [this, cos_pi] at h5
let z := cos (Real.pi / 7)
rcases lt_trichotomy 0 z with hz | hz | hz
any_goals simp_all [show cos (4 * (Real.pi / 7)) = cos (2 * (2 * (Real.pi / 7))) by
  ring,
 cos_two_mul]
any_goals nlinarith [cos_sq_add_sin_sq (Real.pi / 7), Real.sin_pi_div_two_sub,
 pow_two_nonneg (cos (Real.pi / 7) - 1),pow_two_nonneg (cos (Real.pi / 7) + 1)]
contrapose hz
refine not_lt.2 ?_
apply cos_nonneg_of_mem_Icc <;> constructor <;> linarith [pi_pos]
```

**DeepSeek-Prover-V2 (whole-proof):**

```
theorem imo_1963_p5 : Real.cos (Real.pi / 7) - Real.cos (2 * Real.pi / 7) + Real.cos
  (3 * Real.pi / 7) = 1 / 2 := by
 have h1 : 2 * Real.sin (Real.pi / 7) * (Real.cos (Real.pi / 7) - Real.cos (2 * Real.
  pi / 7) + Real.cos (3 * Real.pi / 7)) = Real.sin (Real.pi / 7) := by
   have h1₁ : 2 * Real.sin (Real.pi / 7) * Real.cos (Real.pi / 7) = Real.sin (2 *
  Real.pi / 7) := by
     have h1₁₁ : Real.sin (2 * Real.pi / 7) = 2 * Real.sin (Real.pi / 7) * Real.cos (
  Real.pi / 7) := by
       have h1₁₂ : Real.sin (2 * Real.pi / 7) = Real.sin (2 * (Real.pi / 7)) := by
  ring
       rw [h1₁₂]
       have h1₁₃ : Real.sin (2 * (Real.pi / 7)) = 2 * Real.sin (Real.pi / 7) * Real.
  cos (Real.pi / 7) := by
         rw [Real.sin_two_mul]
         <;> ring
       rw [h1₁₃]
```

```
        <;> ring
      linarith
    have h1₂ : 2 * Real.sin (Real.pi / 7) * Real.cos (2 * Real.pi / 7) = Real.sin (3 *
     Real.pi / 7) - Real.sin (Real.pi / 7) := by

  -- ... (approximately 20 lines of calculations omitted) ...

    have h1₃ : 2 * Real.sin (Real.pi / 7) * Real.cos (3 * Real.pi / 7) = Real.sin (4 *
     Real.pi / 7) - Real.sin (2 * Real.pi / 7) := by

  -- ... (approximately 20 lines of similar calculations omitted) ...

    have h1₄ : Real.sin (4 * Real.pi / 7) = Real.sin (3 * Real.pi / 7) := by

  -- ... (approximately 20 lines of similar calculations omitted) ...

  have h2 : Real.sin (Real.pi / 7) > 0 := by
    apply Real.sin_pos_of_pos_of_lt_pi
    · linarith [Real.pi_pos, Real.pi_gt_three]
    · linarith [Real.pi_pos, Real.pi_gt_three]

  have h3 : Real.cos (Real.pi / 7) - Real.cos (2 * Real.pi / 7) + Real.cos (3 * Real.
   pi / 7) = 1 / 2 := by
    have h3₁ : 2 * Real.sin (Real.pi / 7) > 0 := by linarith
    have h3₂ : Real.cos (Real.pi / 7) - Real.cos (2 * Real.pi / 7) + Real.cos (3 *
   Real.pi / 7) = 1 / 2 := by
      apply mul_left_cancel₀ (show (2 * Real.sin (Real.pi / 7) : ℝ) ≠ 0 by linarith)
      nlinarith [Real.sin_le_one (Real.pi / 7), Real.sin_le_one (2 * Real.pi / 7),
   Real.sin_le_one (3 * Real.pi / 7),
        Real.sin_le_one (4 * Real.pi / 7), Real.sin_le_one (Real.pi / 7)]
    exact h3₂

  apply h3
```

## imo_1963_p5 - Part 2

### Goedel-Prover-V2 (whole-proof):

```
theorem imo_1963_p5 :
  Real.cos (Real.pi / 7) - Real.cos (2 * Real.pi / 7) + Real.cos (3 * Real.pi / 7) = 1
   / 2 := by
  have h1 : 2 * Real.sin (Real.pi / 7) * (Real.cos (Real.pi / 7) - Real.cos (2 * Real.
   pi / 7) + Real.cos (3 * Real.pi / 7)) = Real.sin (Real.pi / 7) := by
    have h1₁ : 2 * Real.sin (Real.pi / 7) * Real.cos (Real.pi / 7) = Real.sin (2 *
   Real.pi / 7) := by
      have h1₁₁ : Real.sin (2 * Real.pi / 7) = 2 * Real.sin (Real.pi / 7) * Real.cos (
   Real.pi / 7) := by
        rw [show Real.sin (2 * Real.pi / 7) = Real.sin (2 * (Real.pi / 7)) by ring,
   Real.sin_two_mul]
        <;> ring
      linarith
    -- ... (approximately 80 lines deriving the product-to-sum identities h1₂, h1₃, h1
   ₄ and combining them through a calc block omitted) ...
  have h2 : Real.sin (Real.pi / 7) > 0 := by
    apply Real.sin_pos_of_pos_of_lt_pi
    <;> linarith [Real.pi_pos, Real.pi_gt_three]
  have h3 : Real.cos (Real.pi / 7) - Real.cos (2 * Real.pi / 7) + Real.cos (3 * Real.
   pi / 7) = 1 / 2 := by
```

```
    apply mul_left_cancel₀ (show (2 : ℝ) * Real.sin (Real.pi / 7) ≠ 0 by linarith)
    nlinarith [h1, Real.sin_le_one (Real.pi / 7), Real.sin_le_one (2 * Real.pi / 7),
      Real.sin_le_one (3 * Real.pi / 7)]
  exact h3
```

**Compfiles dataset (human-proof):**

```
theorem imo1963_p5 :
    Real.cos (π/7) - Real.cos (2*π/7) + Real.cos (3*π/7) = 1/2 := by
  rw [show (2*π/7) = π - (5*π/7) by linarith]
  rw [Real.cos_pi_sub]
  simp only [sub_neg_eq_add]
  have h : 2 * Real.sin (π / 7) ≠ 0 := by
    simp only [ne_eq, mul_eq_zero, OfNat.ofNat_ne_zero, false_or]
    apply ne_of_gt
    apply Real.sin_pos_of_pos_of_lt_pi
    simp only [Nat.ofNat_pos, div_pos_iff_of_pos_right, Real.pi_pos]
    trans 1
    · rw [div_lt_one (by linarith only)]
      linarith only [Real.pi_le_four]
    · linarith only [Real.pi_gt_three]
  apply (mul_right_inj' h).mp
  rw [left_distrib, left_distrib]
  have prod_sum : ∀ (x y : ℝ),
      2 * Real.sin x * Real.cos y = Real.sin (x + y) - Real.sin (y - x) := by
    intro x y
    rw [Real.sin_add, Real.sin_sub]
    linarith only
  rw [prod_sum, prod_sum, prod_sum]
  rw [show (π / 7 + π / 7)     = 2 * π / 7 by linarith only]
  rw [show (π / 7 - π / 7)     = 0         by linarith only]
  rw [show (π / 7 + 5 * π / 7) = 6 * π / 7 by linarith only]
  rw [show (5 * π / 7 - π / 7) = 4 * π / 7 by linarith only]
  rw [show (π / 7 + 3 * π / 7) = 4 * π / 7 by linarith only]
  rw [show (3 * π / 7 - π / 7) = 2 * π / 7 by linarith only]
  rw [Real.sin_zero]
  ring_nf
  rw [← Real.sin_pi_sub]
  rw [show (π - π * (6 / 7)) = π / 7 by linarith]
  congr
  linarith
```

## algebra_amgm_sum1toneqn_prod1tonleq1

For the problem algebra_amgm_sum1toneqn_prod1tonleq1, DeepSeek-Prover-V2 proceeds by manually handling cases ($n = 0$, some $a_i = 0$, all $a_i > 0$), then take the logarithm of the product and apply the well-known inequality $\ln(x) \le x - 1$ to each term, resulting in verbose proofs. In contrast, BFS-Prover-V2 recognizes the problem as a special case of the Arithmetic Mean-Geometric Mean (AM-GM) inequality. It directly invokes the corresponding theorem from Mathlib, Real.geom_mean_le_arith_mean, demonstrating an ability to leverage high-level library theorems for a more insightful and efficient proof.

**BFS-Prover-V2 (step-level proof):**

```
theorem algebra_amgm_sum1toneqn_prod1tonleq1
  (a : ℕ → NNReal)
```

```
  (n : ℕ)
  (h₀ : ∑ x in Finset.range n, a x = n) :
 ∏ x in Finset.range n, a x ≤ 1 := by
 have g := h₀
 revert h₀
 intro amgm
 let S := Finset.range n
 by_cases h1 : n = 0
 simp[h1]
 have hn  : 0 < n := by omega
 let μ := (fun (x : ℕ) => (a x : ℝ))
 let w : ℕ → ℝ := fun _ => 1
 have w_nonneg : ∀ i, 0 ≤ w i
 simp [w, zero_le_one]
 have w_pos  : 0 < ∏ i in Finset.range n, w i
 all_goals aesop (add simp w)
 have w_nonneg : ∀ x ∈ S, 0 ≤ w x
 any_goals simp [w]
 have w_pos  : 0 < ∏ x ∈ S, w x
 simp [w]
 have amgm_real  : (∏ x ∈ S, μ x ^ (w x : ℝ)) ^ (∑ x ∈ S, (w x : ℝ))⁻¹) ≤ ∑ x ∈ S,
   (w x * μ x : ℝ)) / ∑ x ∈ S, (w x : ℝ))
 apply Real.geom_mean_le_arith_mean
 exact w_nonneg
 simp [w, S, hn]
 field_simp[μ]
 simp [μ, w, S] at amgm_real
 norm_cast at amgm_real
 simp[ amgm] at amgm_real
 by_cases H : ∏ a_1 ∈ Finset.range n, a a_1 = 0
 simp [H]
 simp [← NNReal.coe_le_coe, amgm] at amgm_real
 rw  [div_self] at amgm_real
 apply le_of_not_lt
 all_goals aesop (add simp NNReal)
 rw[←NNReal.coe_one] at amgm_real
 norm_cast at amgm_real
 contrapose! amgm_real
 erw [ Real.one_lt_rpow_iff]
 left
 all_goals aesop (add simp [Finset.range])
```

### DeepSeek-Prover-V2 (whole-proof):

```
theorem algebra_amgm_sum1toneqn_prod1tonleq1 (a : ℕ → NNReal) (n : ℕ)
   (h₀ : ∑ x in Finset.range n, a x) = n) : ∏ x in Finset.range n, a x) ≤ 1 := by
 have h₁ : ∏ x in Finset.range n, (a x : ℝ)) ≤ 1 := by
   have h₂ : ∑ x in Finset.range n, (a x : ℝ)) = n := by
     exact_mod_cast h₀
   by_cases h₃ : n = 0
     simp [h₃]
     have h₄ : 0 < n := Nat.pos_of_ne_zero h₃
     by_cases h₅ : ∃ i, i ∈ Finset.range n ∧ (a i : ℝ) = 0
       obtain ⟨i, hi, h₆⟩ := h₅
       have h₇ : ∏ x in Finset.range n, (a x : ℝ)) = 0 := by
         have h₈ : i ∈ Finset.range n := hi
         have h₉ : (a i : ℝ) = 0 := h₆
         have h₁₀ : ∏ x in Finset.range n, (a x : ℝ)) = 0 := by
           calc
             ∏ x in Finset.range n, (a x : ℝ)) = ∏ x in Finset.range n, (a x : ℝ) :=
   rfl
```

```
                  _ = 0 := by
                      apply Finset.prod_eq_zero h₈
                      simp [h₉]
                exact h₁₀
            linarith
          have h₅' : ∀ i, i ∈ Finset.range n → (a i : ℝ) > 0 := by
            intro i hi
            have h₆ : ¬∃ i, i ∈ Finset.range n ∧ (a i : ℝ) = 0 := h₅
            have h₇ : (a i : ℝ) ≠ 0 := by
              intro h
              have h₈ : ∃ i, i ∈ Finset.range n ∧ (a i : ℝ) = 0 := ⟨i, hi, by simp_all⟩
              contradiction
            have h₈ : (a i : ℝ) > 0 := by
              have h₉ : (a i : ℝ) ≥ 0 := by exact_mod_cast (a i).prop
              contrapose! h₇
              linarith
            exact h₈
          have h₆ : (∏ x in Finset.range n, (a x : ℝ)) ≤ 1 := by
            have h₇ : ∑ x in Finset.range n, Real.log ((a x : ℝ)) ≤ 0 := by
              have h₈ : ∑ x in Finset.range n, Real.log ((a x : ℝ)) ≤ ∑ x in Finset.
  range n, ((a x : ℝ) - 1) := by
                apply Finset.sum_le_sum
                intro i hi
                have h₉ : (a i : ℝ) > 0 := h₅' i hi
                have h₁₀ : Real.log ((a i : ℝ)) ≤ (a i : ℝ) - 1 := by
                  have h₁₁ : Real.log ((a i : ℝ)) ≤ (a i : ℝ) - 1 := by
                    linarith [Real.log_le_sub_one_of_pos h₉]
                  exact h₁₁
                exact h₁₀

  -- ... (approximately 40 lines of calculations omitted) ...

  have h₂ : (∏ x in Finset.range n, a x) ≤ 1 := by
    have h₃ : (∏ x in Finset.range n, a x : ℝ) ≤ 1 := by
      exact h₁
    have h₄ : (∏ x in Finset.range n, a x : ℝ) = (∏ x in Finset.range n, a x : ℝ) :=
    rfl
    have h₅ : (∏ x in Finset.range n, a x : ℝ) = (∏ x in Finset.range n, (a x : ℝ)) :=
     by simp
    have h₆ : (∏ x in Finset.range n, a x : ℝ) ≤ 1 := by simpa [h₅] using h₁
    have h₇ : (∏ x in Finset.range n, a x : NNReal) ≤ 1 := by
      norm_cast at h₆ ⊢
      <;> simp_all [Finset.prod_range_succ]
      <;> norm_num
      <;> linarith
    simpa using h₇
  exact h₂
```

**Goedel-Prover-V2 (whole-proof):**

```
theorem algebra_amgm_sum1toneqn_prod1tonleq1
  (a : ℕ → NNReal)
  (n : ℕ)
  (h₀ : ∑ x in Finset.range n, a x = n) :
  ∏ x in Finset.range n, a x ≤ 1 := by
  have h_sum_real : (∑ x in Finset.range n, (a x : ℝ)) = n := by
    norm_cast at h₀ ⊢
    <;> simp_all [Finset.sum_range_succ, NNReal.coe_sum]
  have h_main : (∏ x in Finset.range n, (a x : ℝ)) ≤ 1 := by
    -- ... (approximately 100 lines handling n = 0, the zero-entry case, and the log-
    sum / AM-GM argument for the all-positive case omitted) ...
```

```
  have h_final : ∏ x in Finset.range n, a x ≤ 1 := by
    norm_cast at h_main ⊢ <;> simpa [NNReal.coe_prod] using h_main
  exact h_final
```

Despite the advantages of step-level proof in significantly reducing proof length and discovering novel proof strategies, step-level proof has one notable limitation: poor readability. The interactive nature of step-level proof generation often results in proofs that are more challenging for humans to follow and understand compared to the more structured and explanatory whole-proof approaches. This trade-off between conciseness and readability represents a crucial consideration when evaluating the practical utility of different proof generation paradigms.

## B. Illustration of Planner-Prover Paradigm with an IMO Problem

To demonstrate the effectiveness of our Planner-Prover paradigm, we present an analysis of the solution process for a challenging IMO problem: `imo_1969_p2`.

In the following proof, the statements `h_coeffs_polar`, `h_y_rewritten_with_polar`, and `h_y_collapsed_to_single_cos` represent the dynamic replanning phase, while all other `have` statements belong to the initial planning phase. Unlike in conventional whole-proof methods, `have` statements in our framework are presented without the `:= by` clause. This example highlights the crucial role of dynamic replanning in our system. Without dynamic replanning, the prover gets stuck at `h_y_is_sinusoid`, failing to complete the proof even after 7,200 attempts. With dynamic replanning, however, the system successfully completes the proof in just 800 attempts. The dynamic replanning process breaks down complex steps into smaller, more manageable subgoals, which enables the prover to bypass critical bottlenecks more efficiently.

---

**`imo_1969_p2` - Part 1**

```
theorem imo_1969_p2
  (m n : ℝ)
  (k : ℕ)
  (a : ℕ → ℝ)
  (y : ℝ → ℝ)
  (h₀ : 0 < k)
  (h₁ : ∀ x, y x = ∑ i in Finset.range k, ((Real.cos (a i + x)) / (2^i)))
  (h₂ : y m = 0)
  (h₃ : y n = 0) : ∃ t : ℤ, m - n = t * Real.pi := by
  have h_cos_add : ∀ i x, Real.cos (a i + x) = Real.cos (a i) * Real.cos x - Real.sin
    (a i) * Real.sin x
  simp [cos_add, add_right_inj]

  have h_y_sum_expanded : ∀ (x : ℝ), y x = ∑ i in (Finset.range k : Finset ℕ), (Real.
    cos (a i) * Real.cos x - Real.sin (a i) * Real.sin x) / ((2 : ℕ) ^ i : ℝ)
  simp [h₁, h_cos_add ]

  have h_y_sum_split : ∀ (x : ℝ), y x = ∑ i in (Finset.range k : Finset ℕ), Real.cos
    (a i) * Real.cos x / ((2 : ℕ) ^ i : ℝ)) - ∑ i in (Finset.range k : Finset ℕ),
    Real.sin (a i) * Real.sin x / ((2 : ℕ) ^ i : ℝ))
  intro z <;> simp_rw [h_y_sum_expanded]
  simp [sub_div, Finset.sum_sub_distrib]

  have h_y_expand : ∀ (x : ℝ), y x = ∑ i in (Finset.range k : Finset ℕ), Real.cos (a
    i) / ((2 : ℕ) ^ i : ℝ)) * Real.cos x - ∑ i in (Finset.range k : Finset ℕ), Real.
    sin (a i) / ((2 : ℕ) ^ i : ℝ)) * Real.sin x
  intro x_exp
  simp only [Finset.sum_mul, h_y_sum_split]
  congr <;> symm <;> field_simp <;> ring

  have h_k_ge_one : 1 ≤ k
  apply Nat.succ_le_of_lt    <;> exact h₀

  have h_complex_repr : (∑ i in (Finset.range k : Finset ℕ), Real.cos (a i) / ((2 : ℕ
    ) ^ i : ℝ), ∑ i in (Finset.range k : Finset ℕ), Real.sin (a i) / ((2 : ℕ) ^ i : ℝ)
    ⟩ : ℂ) = ∑ i in (Finset.range k : Finset ℕ), Complex.exp (↑(a i) * Complex.I) / ↑
    (((2 : ℕ) ^ i) : ℝ)
  simp [Complex.exp_mul_I, div_eq_inv_mul, Complex.ext_iff]
  simp [Complex.cos_ofReal_re, Complex.sin_ofReal_re] <;> field_simp <;> norm_cast
  constructor <;> apply Finset.sum_congr <;> aesop
  field_simp [_root_.pow_add, show (4 : ℝ) = 2 ^ 2 by norm_num] <;> ring
  norm_num [mul_comm _ 2, pow_mul]
  rewrite [show (4 : ℝ) ^ x = (2 * 2 : ℝ) ^ x by ring, mul_pow]
    <;> field_simp
    <;> ring
```

```
have h_sum_split : ∑ i in (Finset.range k : Finset ℕ), Complex.exp (↑(a i) *
  Complex.I) / ↑(((2 : ℕ) ^ i) : ℝ)) = Complex.exp (↑(a 0) * Complex.I) + ∑ i in (
  Finset.Icc 1 (k-1) : Finset ℕ), Complex.exp (↑(a i) * Complex.I) / ↑(((2 : ℕ) ^ i)
   : ℝ)
have h_range_split  : Finset.range k = insert 0 (Finset.Icc 1 (k − 1))
ext x  <;>  simp  [Nat.lt_succ_iff]
rcases x with (_|_|x) <;> omega
rw [h_range_split, Finset.sum_insert]
norm_num [pow_zero, eq_self_iff_true]
simp [Nat.le_zero]

have h_abs_head : Complex.abs (Complex.exp (↑(a 0) * Complex.I)) = 1
simp [Complex.abs_exp, eq_self_iff_true]

have h_tail_geom_sum_val : ∑ i in (Finset.Icc 1 (k − 1) : Finset ℕ), 1 / ((2 : ℕ) ^
   i : ℝ)) = 1 − 1 / (2 : ℝ) ^ (k − 1)
have h_tight  : (1 : ℝ) ≤ k
norm_cast at * <;>
linarith
clear h_tight h_sum_split h_complex_repr h_y_expand h_y_sum_split h_y_sum_expanded
  h_cos_add h₂ h₃ h₁ h₀
induction' k <;> simp [Finset.sum_Icc_succ_top, *]
induction' ⟨ℕ⟩ <;> simp_all [Finset.sum_Icc_succ_top, pow_succ]
ring
<;>ring_nf

have h_abs_tail_le : Complex.abs ∑ i in (Finset.Icc 1 (k−1) : Finset ℕ), Complex.
  exp (↑(a i) * Complex.I) / ↑(((2 : ℕ) ^ i) : ℝ)) ≤ 1 − 1 / (2 : ℝ) ^ (k − 1)
rw [← h_tail_geom_sum_val]
apply (Complex.abs.sum_le _ _).trans_eq
apply Finset.sum_congr rfl
intro i _
simp [Complex.abs_exp_ofReal_mul_I, Nat.cast_pow, Nat.cast_ofNat]

have h_abs_tail_lt_one : Complex.abs ∑ i in (Finset.Icc 1 (k−1) : Finset ℕ),
  Complex.exp (↑(a i) * Complex.I) / ↑(((2 : ℕ) ^ i) : ℝ)) < 1
refine lt_of_le_of_lt h_abs_tail_le ?_
refine sub_lt_self _ (by positivity)

have h_abs_ge_by_rev_triangle : Complex.abs ∑ i in (Finset.range k : Finset ℕ),
  Complex.exp (↑(a i) * Complex.I) / ↑(((2 : ℕ) ^ i) : ℝ)) ≥ 1 − Complex.abs ∑ i in
   (Finset.Icc 1 (k−1) : Finset ℕ), Complex.exp (↑(a i) * Complex.I) / ↑(((2 : ℕ) ^
  i) : ℝ))
rw [h_sum_split]
rw [← h_abs_head]
apply Complex.abs.le_add
```

## imo_1969_p2 - Part 2

```
have h_abs_ge_final : Complex.abs ∑ i in (Finset.range k : Finset ℕ), Complex.exp (
  ↑(a i) * Complex.I) / ↑(((2 : ℕ) ^ i) : ℝ)) ≥ 1 / (2 : ℝ) ^ (k−1)
refine' _root_.trans h_abs_ge_by_rev_triangle _
linarith [h_abs_tail_le]

have h_abs_gt_zero : 0 < Complex.abs ∑ i in (Finset.range k : Finset ℕ), Complex.
  exp (↑(a i) * Complex.I) / ↑(((2 : ℕ) ^ i) : ℝ))
linarith [pow_two_nonneg ((k − 1 : ℕ) : ℝ) ]

have h_complex_val_ne_zero : (∑ i in (Finset.range k : Finset ℕ), Real.cos (a i) /
```

```
    ((2 : ℕ) ^ i : ℝ), ∑ i in (Finset.range k : Finset ℕ), Real.sin (a i) / ((2 : ℕ) ^
    i : ℝ)⟩ : ℂ) ≠ 0
  focus all_goals (norm_num; aesop)
```

```
    have h_coeffs_polar : ∃ (R b : ℝ), 0 < R ∧ (∑ i in (Finset.range k : Finset ℕ),
      Real.cos (a i) / ((2 : ℕ) ^ i : ℝ)) = R * Real.cos b ∧ (∑ i in (Finset.range
      k : Finset ℕ), Real.sin (a i) / ((2 : ℕ) ^ i : ℝ)) = R * Real.sin b
    set x := ∑ i ∈ Finset.range k, cos (a i) / ((2 : ℝ) ^ i)
    use Complex.abs (∑ i ∈ Finset.range k, Complex.exp (↑(a i) * Complex.I) / ↑(↑2
      ^ i))
    let y : ℝ := ∑ i ∈ Finset.range k, sin (a i) / 2^i
    have h := Complex.abs_mul_cos_add_sin_mul_I (∑ i in Finset.range k, Complex.exp
      ((a i : ℝ) * Complex.I) / (2 : ℂ) ^ i)
    use Complex.arg (∑ i in Finset.range k, Complex.exp (↑(a i) * Complex.I) / (2:
      ℝ) ^ i)
    simp_all [Complex.ext_iff]
```

```
    have h_y_rewritten_with_polar : ∃ (R a : ℝ), 0 < R ∧ ∀ x, y x = R * Real.cos a
      * Real.cos x - R * Real.sin a * Real.sin x
    obtain ⟨R, phi, hR_pos, h_cos_eq1, h_sin_eq1⟩ := h_coeffs_polar
    use R, phi <;> simp_all[Complex.exp_mul_I, Complex.abs]
```

```
    have h_y_collapsed_to_single_cos : ∃ (R  a : ℝ), 0 < R ∧ ∀ x, y x = R * Real.
      cos (x + a)
    rcases h_y_rewritten_with_polar with ⟨R, a', h_R_pos, h_y_⟩
    use R, a', h_R_pos <;> intros <;> simp [h_y_, cos_add] <;> ring
```

```
  have h_y_is_sinusoid : ∃ (R a : ℝ), 0 < R ∧ (∀ x, y x = R * Real.cos (x - a))
  obtain ⟨ R, a, _, hy ⟩ := h_y_collapsed_to_single_cos
  use R, -a <;> aesop

  have h_roots_exist : ∃ (R a : ℝ), 0 < R ∧ y m = R * Real.cos (m - a) ∧ y n = R *
    Real.cos (n - a)
  rcases h_y_is_sinusoid with ⟨R, a, h_R_pos, h_y_R_a⟩
  exact ⟨ R, a, h_R_pos,
    by simp [h_y_R_a], by simp [h_y_R_a] ⟩

  have h_cos_zero : ∃ (R a : ℝ), 0 < R ∧ Real.cos (m - a) = 0 ∧ Real.cos (n - a) = 0
  rcases h_roots_exist with ⟨R, a, h_rPos,  h_mEq,  h_nEq⟩
  exact
    ⟨R, a, h_rPos,
      by have := h₂; have := h₃; field_simp [h₁] at * <;> nlinarith,
      by have := h₃; have := h₂; field_simp [h₁] at * <;> nlinarith⟩

  have h_roots_in_pi_half_multiples : ∃ (a : ℝ) (t₁ t₂ : ℤ), m - a = (2 * (t₁ : ℝ) +
    1) * Real.pi / 2 ∧ n - a = (2 * (t₂ : ℝ) + 1) * Real.pi / 2
  rcases h_cos_zero with ⟨R, a, _, h_m_cos_zero, h_n_cos_zero⟩
  rw [ cos_eq_zero_iff] at h_m_cos_zero  h_n_cos_zero
  exact ⟨ a,  ↑( Classical.choose h_m_cos_zero ) , ↑( Classical.choose h_n_cos_zero )
    , by convert h_m_cos_zero.choose_spec , by convert  h_n_cos_zero.choose_spec ⟩
```

```
have h_m_minus_n_form : ∃ t₁ t₂ : ℤ, m - n = ((2 * (t₁ : ℝ) + 1) * Real.pi / 2) -
  ((2 * (t₂ : ℝ) + 1) * Real.pi / 2)
obtain ⟨z, t₁, t₂, h_z_root_m, h_z_root_n⟩ := h_roots_in_pi_half_multiples
refine ⟨t₁ , t₂,?_⟩<;>
linarith

have h_m_minus_n_simplified : ∃ t₁ t₂ : ℤ, m - n = (↑(t₁ - t₂) : ℝ) * Real.pi
rcases h_m_minus_n_form with ⟨t₁, t₂, h_form⟩  <;>
  exists t₁  <;> exists t₂  <;>  field_simp at h_form ⊢  <;>  linarith

obtain ⟨t₁, t₂,h_m_sub_n_t₁_t₂⟩ := h_m_minus_n_simplified  <;>  use t₁ - t₂  <;>
  linarith [h_m_sub_n_t₁_t₂]
```

# C. Prompts Used in This Work

## C.1. Prompts for Autoformalization

Our autoformalization pipeline operates in two stages to ensure syntactic correctness. First, an `Initial Formalization Prompt` (shown below) translates a natural language problem into a Lean 4 theorem statement. If the generated code fails to compile, an `Error Feedback Prompt` is then deployed to revise the statement, using the verbatim error message from the Lean compiler as direct feedback for revision.

---

**Prompt for Initial Formalization**

You are an expert in math proof and the theorem prover: Lean. Given a math problem that contains the question and all conditions, and its corresponding solution that contains solution steps and the correct answer, generate a mathematically equivalent proof problem and rewrite it in the Lean 4 statement. You should follow the following procedures.

  a): Identify all questions and conditions in the given problem.

  b): Identify all solution steps and the correct answers in the given solution.

  c): With the questions and conditions in a) and correct answers in b), translate the (question, conditions, correct answer) tuple to a mathematically equivalent proof problem that proves question == answer given conditions.

  d): Rewrite the math proof problem in c) to a Lean 4 statement. Note that you should write the statement only, no proof is required. This also means you do not need to consider the solution steps either.

The first priority is to ensure the generated Lean code can be built successfully. Consider using the following tips.

  • Use a broader import, e.g., `import Mathlib`, to bring in the entirety of the necessary library, and remove specific import of submodules, e.g., `import Mathlib.LinearAlgebra.BasicReal3`, accordingly.

  • Add `noncomputable` before `def` only when necessary.

  • Use `by` instead of `begin end`.

  • Add `sorry` to skip the proof.

You should strictly follow the below criteria to guarantee the lean statement is equivalent to the mathematical problem.

  • Each definition used in Lean 4 statement should only directly appear in the conditions problem in a).

  • Each definition should NOT come from and assume any knowledge directly from the solution step in b).

  • Each condition in a) should be used as a definition in Lean 4.

  • For any implications appearing in the conclusions of the original problem, extract their antecedents and declare them as explicit assumptions before the colon, leaving only the consequent in the conclusion after the colon.

  • For equations, structure the theorem in the form 'conditions : conclusions' where conditions include variable definitions and domains, and conclusions are the solutions to the equation, avoiding implication or equivalence symbols.

**Below are examples to illustrate the process:**

**Example 1 (Number Theory):**
**Lean 4 statement:**

---

```
theorem nt3_problem (n p : ℕ) (hn : n > 1) (hp : Nat.Prime p)
  (h1 : n | (p - 1)) (h2 : p | (n^6 - 1)) :
  ∃ k : ℕ, (p - n = k^2) ∨ (p + n = k^2) := by
  sorry
```

## problem:

NT3. Let $n > 1$ be a positive integer and $p$ a prime number such that $n \mid (p - 1)$ and $p \mid (n^6 - 1)$. Prove that at least one of the numbers $p - n$ and $p + n$ is a perfect square.

### Example 2 (Number Theory):
### Lean 4 statement:

```
theorem nt4_problem (x y : ℕ)
  (hx : x > 0) (hy : y > 0)
  (h1 : ∃ m : ℕ, 3 * x + 4 * y = m^2)
  (h2 : ∃ n : ℕ, 4 * x + 3 * y = n^2) :
  7 | x ∧ 7 | y := by
  sorry
```

## problem:

NT4. If the positive integers $x$ and $y$ are such that both $3x + 4y$ and $4x + 3y$ are perfect squares, prove that both $x$ and $y$ are multiples of 7.

### Example 3 (Algebra):
### Lean 4 statement:

```
theorem sum_not_zero (a b c d : ℝ)
  (h1 : a * b * c - d = 1)
  (h2 : b * c * d - a = 2)
  (h3 : c * d * a - b = 3)
  (h4 : d * a * b - c = -6) :
  a + b + c + d ≠ 0 := by
  sorry
```

## problem:

The real numbers $a, b, c, d$ satisfy simultaneously the equations $abc - d = 1, bcd - a = 2, cda - b = 3, dab - c = -6$. Prove that $a + b + c + d \neq 0$.

### Example 4 (Inequality):
### Lean 4 statement:

```
theorem inequality_proof (a b c : ℝ)
  (ha : a > 0) (hb : b > 0) (hc : c > 0) :
  8 / ((a + b)^2 + 4*a*b*c) +
  8 / ((b + c)^2 + 4*a*b*c) +
  8 / ((c + a)^2 + 4*a*b*c) +
  a^2 + b^2 + c^2 ≥
  8 / (a + 3) + 8 / (b + 3) + 8 / (c + 3) := by
  sorry
```

## problem:

The real numbers $a, b, c, d$ satisfy simultaneously the equations $abc - d = 1, bcd - a = 2, cda - b = 3, dab - c = -6$. Prove that $a + b + c + d \neq 0$.

Now, use the same process for the following problem and solution:

{**problem**}

{**solution**}

## Prompt for Error Feedback

You are an expert in math proof and the theorem prover: Lean. You are given the following math problem that contains the question and all conditions, and its corresponding solution that contains solution steps and the correct answer.

{**problem**}

{**solution**}

A mathematically equivalent proof problem that proves question == answer given conditions is generated and rewritten in the Lean 4 statement, as shown below:

{**Lean 4 statement**}

However, this lean code got error with `lake build`, and here is the error message:

{**error message**}

Please modify the lean code to ensure it can be built successfully with `lake build`. Here is a few tips that might help:

- Use a broader import, e.g., `import Mathlib`, to bring in the entirety of the necessary library, and remove specific import of submodules, e.g., `import Mathlib.LinearAlgebra.BasicReal3`, accordingly.

- Add `noncomputable` before `def` only when necessary.

- Use `by` instead of `begin end`.

- Add `sorry` to skip the proof.

### C.2. Prompts for Planner

## Prompt for Initial Planning

You are an expert assistant specializing in Math Olympiads and the Lean 4 theorem prover. Your primary goal is to generate **syntactically perfect, type-checkable** Lean 4 intermediate step code snippets (**plan**) for a given theorem. It is crucial to strictly adhere to the following rules—any violation will be considered an error.

**Task**
Given the following Lean 4 theorem tactic state, generate the core intermediate subgoals (`have` statements) needed for the proof.

**Mandatory Rules**
You must comply with every rule in this section. Failure to adhere to any single rule will result in an incorrect output.

1. **Critical Rule: Explicitly Specify Set/Finset Types**
   This is the most common and fatal point of error. You must explicitly declare the type for any `Set` or `Finset` literal. This rule is non-negotiable.

   ```
   - Correct: ({ {-1, 0, 1}} : Set ℤ)
   - Incorrect: { {-1, 0, 1}}
   ```

2. **Omit the Proof**: Never provide the proof. Only state the `have` statement itself.

3. **Valid Lean 4 Code**: The entire output block must be type-checkable in a Lean 4.10.0 environment.

4. **Use Existing Names**: Use the exact, existing lemma and definition names from `Mathlib`. Do not invent names.

5. **No Undeclared Variables**: Do not introduce any variables or constants not declared in the original theorem statement.

6. **Explicit Multiplication**: Multiplication must always use the ∗ symbol.

   – Correct: `a * x`
   – Incorrect: `ax`

7. **No Chained Inequalities**: Never use chained inequalities. They must be split using logical AND ∧.

   – Correct: `a <= x ∧ x <= b`
   – Incorrect: `a <= x <= b`

8. **Correct Logarithm Function**: `Real.log` is only for the natural logarithm. For logarithms with a specified base, you must use `Real.logb`.

   – Correct: `Real.logb (2 : ℝ) 8`
   – Incorrect: `Real.log (2 : ℝ) 8`

9. **Factorial Notation**: In Lean, factorials must be written as `(n)!` or `Nat.factorial n`, not `n!`.

   – Correct: `(n)!` or `Nat.factorial n`
   – Incorrect: `n!`

10. **Numeric Types Must Be Explicitly Annotated**: To avoid type ambiguity in Lean, any expression involving numeric operations must have at least one number's type specified.

    – For division: `(1 : ℝ) / 2 = 0.5`, but `(1 : ℤ) / 2 = 0`.
    – For subtraction: `(1 : ℤ) - 2 = -1`, but `(1 : ℕ) - 2 = 0`.
    – Correct: `(a : ℝ) / b, a / (b : ℝ), (n : ℤ) - m`
    – Incorrect: `a / b, n - m`

11. **Interval Notation**: Do not use `Icc`, `Ioo`, `Ico`, `Ioc`, etc., to represent intervals. Only use inequalities.

    – Correct: `a <= x ∧ x <= b`
    – Incorrect: `Icc a b`

12. **Complex Numbers**: Use `Complex.I` for the imaginary unit and `Complex.abs` for the modulus/absolute value of a complex number.

13. **Avoid Common Inequality Theorems**: Avoid using common inequality theorems like Holder's or Jensen's. For inequality problems, try to ensure each proof step only requires basic simplification.

14. **Proving Equivalences**: For proofs of equivalences (iff), ensure each `have` statement is an implication, where the antecedent is the left side of the equivalence (when proving left-to-right) or the right side (when proving right-to-left).

15. **Real.pi Notation**: Consistently use `Real.pi` instead of $\pi$.

16. **Final Check**: Before providing the plan, perform a final review to ensure you have scrupulously followed all the rules above, especially the critical rule regarding `Set`/`Finset`.

**Below are examples to illustrate the process:**

**Example 1:**
**Theorem:**

```
theorem singapore2019_r1_p7 (x : ℝ) (hx : Real.tan x = 5) :
  (6 + Real.sin (2 * x)) / (1 + Real.cos (2 * x)) = 83 := by
```

**Plan:**

```
have h₁ : Real.sin x = 5 * Real.cos x
have h₂ : Real.sin x ^ 2 = 25 * Real.cos x ^ 2
have h₃ : 26 * Real.cos x ^ 2 = 1
have hsin2x_val : Real.sin (2 * x) = (5 : ℝ) / (13 : ℝ)
have hcos2x_val : Real.cos (2 * x) = -(12 : ℝ) / (13 : ℝ)
```

### Example 2:
**Theorem:**

```
theorem problem4
  (g : ℕ → ℝ)
  (h : ∀ k : ℕ, 5 ≤ k → k ≤ 124 → g k = (Real.logb (k : ℝ) ((7 : ℝ) ^ (k ^ 2 - 1))) /
    (Real.logb ((k + 1 : ℝ)) ((7 : ℝ) ^ (k ^ 2 - 4)))) :
  (∏ k in Finset.Icc (5 : ℕ) 124, g k) = (41 : ℝ) / 7 := by
```

**Plan:**

```
have h_prod_split : (∏ k in (Finset.Icc 5 124 : Finset ℕ), g k) = (∏ k in (Finset.
  Icc 5 124 : Finset ℕ), ((k ^ 2 - 1) / (k ^ 2 - 4 : ℝ))) * (∏ k in (Finset.Icc 5
  124 : Finset ℕ), (Real.logb (k : ℝ) (7 : ℝ) / Real.logb ((k + 1 : ℝ)) (7 : ℝ)))
have h_telescope_part1 : (∏ k in (Finset.Icc 5 124 : Finset ℕ), ((k ^ 2 - 1) / (k ^
  2 - 4 : ℝ))) = (41 : ℝ) / 21
have h_telescope_part2 : (∏ k in (Finset.Icc 5 124 : Finset ℕ), (Real.logb (k : ℝ)
  (7 : ℝ) / Real.logb ((k + 1 : ℝ)) (7 : ℝ))) = 3
have h_final_product : (41 / 21 : ℝ) * 3 = (41 : ℝ) / 7
```

### Example 3:
**Theorem:**

```
theorem amc12b_variant_p13
  (S : Finset ℝ)
  (h₀ : ∀ (x : ℝ), x ∈ S ↔ 0 < x ∧ x ≤ 2 * Real.pi ∧ 2 - 4 * Real.sin x + 3 * Real.cos
    (3 * x) = 0) :
  S.card = 4 := by
```

**Plan:**

```
have h_interval1 : ∃ x, 0 ≤ x ∧ x < Real.pi / 2 ∧ (2 - 4 * Real.sin x + 3 * Real.cos
  (3 * x) = 0)
have h_interval2 : ∃ x, Real.pi / 2 ≤ x ∧ x < 3 * Real.pi / 4 ∧ (2 - 4 * Real.sin x
  + 3 * Real.cos (3 * x) = 0)
have h_interval3 : ∃ x, 3 * Real.pi / 4 ≤ x ∧ x < Real.pi ∧ (2 - 4 * Real.sin x + 3
  * Real.cos (3 * x) = 0)
have h_interval4 : ∃ x, Real.pi ≤ x ∧ x < 2 * Real.pi ∧ (2 - 4 * Real.sin x + 3 *
  Real.cos (3 * x) = 0)
have h_card_eq_4 : S.card = 4
```

Now, use the same process for the following theorem:

## {theorem}

You must follow all the mandatory rules above. After deep consideration, provide the Lean 4 intermediate step code snippets. While ensuring correctness, the more intermediate steps, the better.

## Prompt for Dynamic Replanning

You are an expert assistant specializing in Math Olympiads and the Lean 4 theorem prover, with a particular talent for proof decomposition and overcoming difficult proof steps.

Your primary goal is to refine an existing proof plan by inserting more granular, logically sound subgoals to help a prover overcome a specific, identified bottleneck.

It is crucial to strictly adhere to the following rules—any violation will be considered an error.

**Task**
Given a Lean 4 theorem, its initial proof plan, and a specific `have` statement where a prover has become stuck, your task is to generate a new, **complete proof plan**.

This new plan must include all the original steps, but with additional, simpler `have` statements inserted **immediately before** the 'stuck' subgoal. These new steps must logically lead to the proof of the stuck subgoal, breaking down the complex reasoning into a series of more manageable steps.

**Mandatory Rules**
You must comply with every rule in this section. Failure to adhere to any single rule will result in an incorrect output.

(The first 16 rules are identical to those in the `Prompt for Initial Planning` and must be strictly followed.) In addition, the following task-specific rules apply:

17. **Insert Before Stuck Step**: The new auxiliary `have` statements must be inserted **immediately before** the provided 'stuck' subgoal.

18. **Provide Complete Plan**: The output must be the **entire, updated plan**, including all original and new `have` statements in the correct order. Do not output only the new steps.

19. **Logical Progression**: The newly inserted steps must be logically sound and serve as direct prerequisites for proving the stuck subgoal. They should bridge the logical gap that caused the prover to get stuck.

**Below is an example to illustrate the process:**

**Theorem:**

```
theorem trig_identity_4x (x : ℝ) :
  Real.sin (4 * x) = 4 * Real.sin x * Real.cos x * (1 - 2 * Real.sin x ^ 2) := by
```

**Initial Plan:**

```
  have h_sin4x_is_2sin2xcos2x : Real.sin (4 * x) = 2 * Real.sin (2 * x) * Real.cos (2
    * x)
  have h_final_identity : 2 * Real.sin (2 * x) * Real.cos (2 * x) = 4 * Real.sin x *
    Real.cos x * (1 - 2 * Real.sin x ^ 2)
```

**Stuck Subgoal:**

```
  have h_final_identity : 2 * Real.sin (2 * x) * Real.cos (2 * x) = 4 * Real.sin x *
    Real.cos x * (1 - 2 * Real.sin x ^ 2)
```

**Refined Plan (Your Output):**

```
  have h_sin4x_is_2sin2xcos2x : Real.sin (4 * x) = 2 * Real.sin (2 * x) * Real.cos (2
    * x)
  have h_sin2x : Real.sin (2 * x) = 2 * Real.sin x * Real.cos x
  have h_cos2x_in_terms_of_sin_cos : Real.cos (2 * x) = Real.cos x ^ 2 - Real.sin x ^
    2
```

```
  have h_cos2x_in_terms_of_sin : Real.cos (2 * x) = 1 - 2 * Real.sin x ^ 2
  have h_final_identity : 2 * Real.sin (2 * x) * Real.cos (2 * x) = 4 * Real.sin x *
    Real.cos x * (1 - 2 * Real.sin x ^ 2)
```

Now, use the same process for the following three items:

{**theorem**}

{**initial_plan**}

{**stuck_subgoal**}

You must follow all the instructions and mandatory rules above. After deep consideration, provide the complete, refined Lean 4 plan.

