# OpenReview forum: "Scaling up Multi-Turn Off-Policy RL and Multi-Agent Tree Search for LLM Step-Provers"
_ICML.cc/2026/Conference — ICML 2026 regular_

### Official Review · Reviewer_7rcS · 2026-03-10

**Soundness:** 3
**Presentation:** 3
**Significance:** 2
**Originality:** 2
**Overall Recommendation:** 5
**Confidence:** 4

**Summary:**

The paper builds on top of BFS-Prover-V1, enhancing both the training and the inference scheme, and achieving very strong performance on MiniF2F and ProofNet.

Training is performed using expert iteration similarly to GPT-f. However, instead of re-generating the whole training dataset at each iteration, it behaves in an accumulative way, akin to a replay buffer reminiscent to AlphaZero.
Additionally, the method present three training improvements:
- The state+tactic training pairs are filtered by their predicted logprob so as to discard tactics that are too simple or too complex.
- When the training plateaus in terms of MiniF2F performance, the whole training dataset is re-generated from scratch.
- When the training plateaus for the 7B model, it's switched for the 32B model. This is a clever way of bootstrapping the expert iteration quickly before transitioning to a stronger model.
All three improvements are shown to have positive effect in the performed training run.

At test time, a separate general-purpose LLM is optionally utilized as a planner, decomposing the proof into subgoals that are then proven using BFS-Prover. When a subgoal cannot be proven, the general-purpose LLM also handles replanning. Authors utilize Gemini 2.5 Pro as the planner.
Further, when attempting to prove a subgoal, several parallel provers attempt the proof search independently, terminating once a solution is found or compute budget runs out.

**Compliance With Llm Reviewing Policy:**

Affirmed.

**Final Justification:**

The detailed rebuttal, in particular the authors' commitment to making certain edits in the paper and to open-sourcing their method, has largely addressed my concerns. I believe this is a strong contribution to the automated theorem proving research. I raise my recommendation to Accept.

**Key Questions For Authors:**

1. Do you plan to release the training and inference source code or the autoformalized dataset as open-source? Open-sourcing these important artifacts would enable reproducibility and building on top of BFS-Prover-V2, and would influence my scoring on the Significance front.
2. How was the set of formal statements assembled? What is the role of the cited datasets LEAN-Github, Lean-WorkBook, and Mathlib? What is the ratio of different data sources in the statements set?
3. How were the human-authored training data obtained? Were any measures taken to asses or mitigate test-set contamination?

**Limitations:**

Limitations of the presented approach are not discussed.

**Strengths And Weaknesses:**

*Note on notation*: By [104l] and [048r], I mean line 104 left and line 48 right, respectively.

**Soundness:**

The presented method is demonstrated to have strong performance on relevant benchmarks and the training run is described in detail.

Areas of improvement:
- It's unclear whether any measures were taken to assess or mitigate test set contamination in the training corpus. The origin of the human-authored data is not explained.
- Ideally, the three presented training improvements should be evaluated more thoroughly than only based on anecdotal evidence from one training run. However, even now, the anecdotal justification is meaningful and useful.
- The paper should include information about compute requirements needed for the training run, and potentially for all the ablations.
- Sections 2.2.1 and 2.2.2 should state the exact portion of discarded data in the low and high tails of the logprob distribution. Specifically, in Section 2.2.2, this concerns the "Aggressive Data Curation" step [242l].
- [324r] It should be noted which LLMs were used for the autoformalization.
- Section 2.3.2: A potential valuable ablation could explore whether some form of work sharing during multi-agent collaboration could be viable. Right now, all parallel workers run BFS proof search on the same subgoal without any inter-prover communication, which intuitively seems wasteful and should be investigated.
- Section 2.3.2: Another ablation could investigate simple heuristic approaches to selecting the order in which to explore the subgoals, to foster the fail-fast approach mentioned in [325l]. For example based on subgoal size in terms of number of characters (longest to shortest).
- The Budget column of Table 1 is not explained or addressed anywhere in the paper and is impossible to interpret without additional context.

**Presentation:**

The paper is well structured and easy to follow.

Areas of improvement:
- Section 2.3.2: It should be explained how provers are randomized, i.e. how the independence of searches are achieved, since the best-first search algorithm is deterministic.
- [363r] mentions human data, but the paper does not explain what they are. Are these data extracted from Mathlib? If so, what method was used for the extraction?
- The appendix should include a table listing training hyperparameter values. Most importantly, the paper doesn't mention how many proofs or state-tactic pairs are collected in each expert iteration and how they are sampled for training. We only have partial information: [133r] mentions that 10 million searches are performed in each iteration, but not how many proofs are collected. Then, [325r] mentions that the number of available formal statements is 3 million. This additionally raises the question of how statements are sampled for the provers, since there are more searches in each iteration than statements available.
- [373l] "We attribute this superior OOD performance to the inherent flexibility of step-level proving: ... a trained step-prover can adapt its exploration strategy by adjusting search parameters at test time to match problem distributions, enabling effective transfer without retraining" - This implies that the hyperparameter setting was different on ProofNet than on MiniF2F. How were these settings chosen?
- [326l] The grid search configuration should be mentioned - i.e. which hyperparameter configurations were tried.
- [317l] It's unclear whether the training starts from plain Qwen2.5 or from an already trained BFS-Prover-V1. From the results I assume the latter, but it should be clarified.
- It should be explained how it was decided when to trigger the retrain and scale-up steps. Was it done manually or according to some automatic heuristic?
- [327r] It's unclear why LEAN-Github, Lean-WorkBook and Mathlib are cited in this context. The role of Goedel-Pset-V1 is stated, but what is the role of these three other data sources? Were they utilized during training?

**Significance:**

The paper presents valuable contributions to the formal theorem proving community, mainly the adaptive tactic filtering, soft resets, and strong ProofNet performance.

However, the replication and building on top of the presented method are hindered by the omission of critical details in the paper, and by the fact that authors don't release the training and inference code together with the autoformalized dataset.

**Originality:**

The paper doesn't include a section on related work and fails to acknowledge existing publications critically relevant to the presented method. Critical omissions include:
- GPT-f (Polu & Sutskever, 2020) - The first application of expert iteration in formal theorem proving.
- HyperTree Proof Search (Lample et al., 2022) and AlphaProof (Hubert et al., 2025) - Methods utilizing variants of Monte Carlo Tree Search, inspired by AlphaZero. The presented paper mentions AlphaZero inspiration [152l], but fails to cite existing publications that extend AlphaZero to formal theorem proving.

Additionally, Draft, Sketch, and Prove (Jiang et al., 2022) is cited but its relation to the presented paper should be discussed in more detail as the DSP approach closely resembles the planner introduced in Section 2.3.

---

> ### Author Rebuttal · Authors · 2026-03-31
>
> We thank the reviewer for the detailed review and address each concern below.
> ## Soundness
> - Data contamination: We applied (1) exact and fuzzy string matching against all miniF2F and ProofNet statements to exclude them from training; (2) source-level separation: training statements come from autoformalized NuminaMath and Goedel-Pset-V1, stylistically distinct from miniF2F (competition) and ProofNet (textbook); (3) empirical validation: strong OOD generalization on ProofNet (41.4%, surpassing 671B models) and DeepSeek-ProverBench (3.08) indicates genuine reasoning. "Human data" = state-tactic pairs from Mathlib v4.10 via LeanDojo (459,540 pairs). We will clarify this terminology.
> - Ablation depth: Fully controlled cross-run ablations would be ideal but infeasible (a single run = 22 expert iteration rounds across 7B and 32B). We provide: (1) within-run ablations with quantitative measurements; (2) training dynamics in Figure 4 showing clear plateaus before each intervention and gains afterward; (3) replicated evidence across multiple plateau-recovery events. Cross-run ablations would require multiple full pipelines costing thousands of A100 GPU hours each. We will note this as a limitation.
> - Computational cost: Full-parameter SFT for 7B ~250-300 A100 GPU hours/epoch, 32B ~1,500-1,800 A100 GPU hours/epoch. Each expert iteration round ~1 epoch (retraining from base: 3 epochs). Budget breakdown will be in the revision.
> - Filtering thresholds: During restart phases, we discard top and bottom ~20% of the log-probability distribution; during non-restart iterations, ~15%. Exact values will be added to Sections 2.2.
> - LLMs for autoformalization: GPT-4o and Claude 3.7 Sonnet (depending on API credits). Will be specified.
> - Multi-agent work sharing: Parallel provers on the same subgoal run independent BFS without inter-prover communication. The shared subgoal cache already provides collaboration: once any prover solves a subgoal, all others immediately benefit. Intra-search parallelism is a promising future direction.
> - Subgoal ordering: Currently, planner-generated subgoals have logical dependencies constraining order. Assuming earlier subgoals (via sorry or local hypotheses) to attempt later ones in parallel could improve efficiency; we will include this as future work.
> - Budget column in Table 1: For whole-proof methods, budget is pass@K; for tree-search methods, (passes x branching factor x instance lifetime). E.g. BFS-Prover-V1's "2048x2x600" = 2048 passes, branching factor 2, 600s lifetime each. We will add an explicit explanation.
> ## Presentation
> - BFS randomization: BFS itself is deterministic; the randomness comes from LLM sampling. We use a temperature of 1.3, so the decoded output varies. We will add this clarification.
> - "Human data": See Soundness above.
> - Hyperparameter selection: Via grid search on a held-out validation set. Different benchmarks have structurally different problems (Olympiad vs. textbook), leading to different optimal configurations.
> - Grid search: branching factors 2,3,4,8; depth rewards 0,1,2.
> - Base checkpoint: All training starts from Qwen2.5-Math-7B or Qwen2.5-32B. Checkpoint 0 in Figure 4 = BFS-Prover-V1 (fine-tuned from Qwen2.5-Math-7B). Will be clarified in the figure caption.
> - Retrain/scale-up trigger: Determined manually by (1) newly solved problems dropping to near zero, (2) downstream performance stagnating or declining.
> - Citation context: LEAN-Github and Lean-WorkBook were cited in wrong context; they are not used in training. Goedel-Pset-V1 was used. Will be corrected.
> ## Significance
> We will release model weights (7B and 32B), inference code (already at https://anonymous.4open.science/r/BFS-Prover-V2-365E), and autoformalized data after decontamination. Training uses standard full-parameter SFT identical to any open-source implementation.
> ## Originality
> We will make the following additions:
> - GPT-f: We will add a formal citation and acknowledge it as the first application of expert iteration in formal theorem proving.
> - HyperTree Proof Search and AlphaProof: Both are already cited ([042R, 449R] for HTPS; [152L, 506L] for AlphaZero). We will make the connection to our work more explicit.
> - Draft, Sketch, and Prove: We will expand the discussion to more explicitly acknowledge how DSP's approach inspired our planner design.
> ## Questions
> 1. See Significance above. We will release weights, inference code, and data.
> 2. Training data: \~3M formal statements, Goedel-Pset-V1 (\~1.73M, \~58%) and self-autoformalized from NuminaMath (\~1.27M, \~42%). Our cold-start data uses Mathlib v4.10 exclusively. Composition table will be in revision.
> 3. Test-set contamination: As detailed in Soundness above (exact/fuzzy matching, source-level separation, empirical OOD validation).
> ## Limitations
> We will add: (1) Planner as external dependency not co-trained with prover. (2) Lack of intra-search parallelism among parallel provers. (3) Subgoal ordering constrained by logical dependencies.

---

> > ### Author Rebuttal · Reviewer_7rcS · 2026-04-02
> >
> > Thank you very much for your detailed rebuttal and for your contribution. Your commitment to making certain edits and to open-sourcing your approach resolved most of my concerns.
> >
> > Regarding AlphaProof, lines [152L, 506L] only cite AlphaZero. I still believe citing AlphaProof is critical, as it is the current SOTA in tree-search-based automated theorem proving on MiniF2F.

---

> > > ### Author Response · Authors · 2026-04-04
> > >
> > > Thank you for the follow-up. Apologies for the misreading. You are correct that [152L, 506L] cite AlphaZero, not AlphaProof. We will add an explicit citation of AlphaProof (Hubert et al., 2025) in the revision and properly discuss it as a key work that extends AlphaZero-style methods to formal theorem proving.
> > >
> > > We hope this fully resolves your remaining concern. With all raised issues addressed, we would be very grateful if you could consider raising your score accordingly.

---

### Official Review · Reviewer_gnrk · 2026-03-13

**Soundness:** 3
**Presentation:** 4
**Significance:** 4
**Originality:** 4
**Overall Recommendation:** 5
**Confidence:** 4

**Summary:**

This paper targets two fundamental problems in automated theorem proving: performance plateau in expert iteration, and inefficient exploration in search space. It combines a generalist LLM planner for proof decomposition, a specialist LLM prover, and uses subgoal cache and dynamic replanning mechanisms to improve efficiency in proof search. For LLM prover training, it proposes vast datasets with perplexity-based training data filtering and periodic retraining to alleviate performance plateau in traditional expert iteration loop.

**Compliance With Llm Reviewing Policy:**

Affirmed.

**Final Justification:**

All weakenesses in "Strengths And Weaknesses" are satisfactorily addressed.

**Key Questions For Authors:**

(Identical to "major weaknesses (W)" in "Strengths And Weaknesses")

**Limitations:**

No. It seems that this paper lacks a Limitation section.
In my humble opinion, limitations may include:
1. Computational resource requirement. It is extremely hard for the community to reproduce or follow their training and even inference (Pass@8192)
2. Data efficiency. Current pipeline relies on Expert iteration and adaptive data filtering. Expert iteration drops all negative samples, and data filtering drops all tactics with high or low perplexity.

**Strengths And Weaknesses:**

Overall, I think the strengths (S) of this paper outweigh its weaknesses (W).

S1. (Originality & Soundness) It proposes two reasonable and generalizable techniques to alleviate performance plateau in expert iteration. Extensive experiments of 22 rounds validates their effectiveness.

S2. (Originality & Significance) It represents a revival of the proof search paradigm and proposed a framework to reach nearly SOTA performance compared with whole-proof generation methods. (Although the pipeline is not pure proof search but a mixed high-level decomposition + low-level proof search manner).

S3. (Presentation) The overall framework is well-presented and easy to follow.

S4. (Soundness) The proposed components are validated by comprehensive ablations.

However, I found several major weaknesses (W) and minor weaknesses (w) worth noting. Actively resolving them can facilitate future readers' understanding, help the community to construct reusable, generalizable knowledge, and as the authors suggest, "may be applied to other domains". I totally understand some of them are computationally expensive and the rebuttal time is short. Therefore, not resolving minor points will NOT lead to lowering my scores.

W1. (Presentation) What does "accumulative" mean in Table 1? Sec. 3.4 says "accumulative performance across a small grid of search hyperparameters (varying branching factors and depth rewards) with a budget of pass@8192 per configuration". What are the "small grid of search hyperparameters"? Does "accumulative" mean pass@(N * 8192), where N is the size of the "small grid"?

W2. (Soundness) It seems that results in Table 1 does not share an aligned computational budget. Could the author provide a fairer comparison? For example, reporting both pass@32 and the count of output tokens across methods, or reporting pass@K when budgets are aligned?

W3. (Presentation) How do you define "Pass@K" for BFS-Prover-V2? Does it take the planner's decomposed lemma and dynamic replanning into account? Does cache hit viewed as zero-cost? Additional clarification would be helpful?

w1. (Presentation) Slightly overclaim regarding "state-of-the-art results on established formal mathematics benchmarks" in abstract. To the best of my understanding, agentic pipelines such as Seed Prover, Delta Prover and Hilbert Prover demonstrate higher performance. However, this is not a significant mistake. I really appreciate that this paper pushes the frontier of proof-search methods and open-sourced models.

w2 (Soundness) Current evaluation omits more difficult undergrad-level and PhD-level math benchmarks including PutnamBench and FATE. How will BFS-Prover-V2 perform on these benchmarks?

w3 (Soundness) Current qualitative ablation study on Subgoal cache with merely two empirical examples not convincing. Could the authors provide a more quantitative result?

w4. (Soundness). Sec. 3.4 reports that "a single fixed configuration (branching factor 3 with depth reward 2 for miniF2F, or branching factor 8 with depth reward 1 for ProofNet) yields comparable results given a budget of pass@16384". Could the author please clarify why different hyperparameters are used or fix their hyperparameters to provide a generalizable setting across benchmarks to facilitate community use?

w5. (Presentation) Sec. 3.3, "Perplexity-based tactic filtering", "The training corpus consisted of 459,540 pairs of human data". Clarification is needed regarding 'human data'. Do they refer to successful statement-proof pairs?

w6. (Soundness): The authors claim the ppl of tactics roughly follows a Gaussian distribution, but Fig. 2 is asymmetric and looks more like a long-tailed Gamma distribution.

[1] Hilbert: Recursively Building Formal Proofs with Informal Reasoning

[2] FATE: A Formal Benchmark Series for Frontier Algebra of Multiple Difficulty Levels

---

> ### Author Rebuttal · Authors · 2026-03-31
>
> We thank the reviewer for the detailed review and recognition of our work's originality and significance. We address each concern below.
> ## Major Weaknesses
> W1: "Accumulative" means the union of problems solved across a small grid of search hyperparameters, with pass@8192 per configuration. We searched over branching factors 2,3,4,8 and depth rewards 0,1,2. For miniF2F, the union of (branching 2, depth reward 2) and (branching 4, depth reward 1) reproduces the accumulative result, though the best single configuration is (branching 3, depth reward 2). For ProofNet, the best is (branching 8, depth reward 1). The gap between the best single configuration and the accumulative result is only 1-2 problems. We report accumulative results to rigorously estimate the system's maximum reasoning capability. We will make this definition more explicit in the revision.
>
> W2: We will add pass@8192 results under a single fixed configuration for each benchmark in Table 1. For BFS-Prover-V2-32B, pure tree search achieves 85.25% on miniF2F @8192 × 3 × 600 and 40.32% on ProofNet under 8192 × 8 × 600, both with a 600s wall-time timeout per instance. Our primary baselines (DeepSeek-Prover-V2-671B, Goedel-Prover-V2-32B, Delta-Prover) all report pass@8192 with accumulative metrics.
>
> W3: Pass@K definition: Without planner, K independent tree searches. With planner, K cumulative prover instances; each contributes to the shared subgoal cache. When a prover solves a subgoal, subsequent instances skip it via cache replay. Cache hits are pure CPU operations with negligible overhead, and can be treated as approximately zero-cost. We will add this clarification in the revision.
> ## Minor Weaknesses
> w1: We will revise claims to: "state-of-the-art results among tree-search provers on established formal mathematics benchmarks, and competitive with the best whole-proof and agentic methods." We fully acknowledge the strong results from Seed-Prover, Delta-Prover, and Hilbert Prover, and will position our contribution more carefully as pushing the frontier of the tree-search paradigm specifically. On ProofNet, BFS-Prover-V2 achieves the best reported result among open-source provers and also surpasses existing whole-proof baselines.
>
> w2: We appreciate this suggestion. We have conducted additional evaluations:
>
> |DeepSeek-ProverBench|Score|
> |---|---|
> |DeepSeek-Prover-V2-7B|0.31|
> |Kimina-Prover-Distill-8B|1.38|
> |Kimina-Prover-72B|1.53|
> |Goedel-Prover-V2-8B|1.53|
> |Goedel-Prover-V2-32B|1.85|
> |Goedel-Prover-V2-32B (w/ revision)|2.46|
> |BFS-Prover-V2-32B|1.85|
> |BFS-Prover-V2-32B (w/ planner)|3.08|
>
> |ProofNet|Score|
> |---|---|
> |DeepSeek-Prover-V2-7B|23.0|
> |Goedel-Prover-V2-32B|22.6|
> |Goedel-Prover-V2-32B (w/ RAG)|28.5|
> |DeepSeek-Prover-V2-671B|37.1|
> |BFS-Prover-V2-32B|41.4|
>
> (data partially from Goedel-Prover-V2's OpenReview comments)
>
> FATE currently lacks a version compatible with our Lean setup. PutnamBench contains numerous errors on older Lean versions and requires substantial compute for full evaluation. We will make every effort to include these in the revision.
>
> w3: We agree two examples are insufficient and are conducting a full evaluation. Preliminary observations: many problems require >8192 passes without cache but only ≤512 cumulative instances with it, including `algebra_sum1onsqrt2to1onsqrt10000lt198`, `imo_1969_p2`, `imo_1965_p2`, etc. Theoretically, for n subgoals each requiring k passes: without cache O(k^n), with cache O(n·k), an exponential-to-linear improvement.
>
> w4: Optimal hyperparameters differ because miniF2F problems are Olympiad-style requiring deeper tactic chains (smaller branching factor 3, higher depth reward 2), while ProofNet problems are undergraduate textbook-style solvable with broader shallower search (larger branching factor 8, lower depth reward 1). This is a strength of step-level provers: unlike whole-proof generation where style is fixed, search hyperparameters adapt to different problem distributions without retraining. Configurations were selected via grid search on a held-out validation set.
>
> w5: "Human data" refers to state-tactic pairs extracted from Mathlibv4.10 using LeanDojo. We will clarify this in the revision.
>
> w6: We thank the reviewer for the excellent observation. We agree the distribution is right-skewed and more accurately Gamma-like. "Roughly Gaussian" was informal to convey unimodality with identifiable tails. The key insight that filtering tails improves training quality is distribution-agnostic. We will correct to "roughly Gamma distribution"
> ## Limitations
> We will add a Limitations section covering:
> 1. Computational cost: SFT for 7B ~250-300 A100 hours/epoch, 32B ~1500-1800 A100 hours/epoch. We note that studying how performance scales with compute is one of our goals; the substantial usage is part of the scientific question, not merely a byproduct.
> 2. Data efficiency: the pipeline discards negative samples and aggressively filters tactics. Incorporating negative feedback is a promising direction.

---

> > ### Author Rebuttal · Reviewer_gnrk · 2026-04-03
> >
> > Thank you authors for the comprehensive response. All the above weaknesses (W1-W3, w1-w6) are satisfactorily addressed. I will raise the "Presentation" score to 4 and the overall score to 5.
> >
> > BTW. what do the scores for `DeepSeek-ProverBench` in response to w2 represent? The original paper [1]'s Table 6, "Experimental Results on ProverBench", reports pass@k values ranging from approximately 27.5% to 59.1%. Are your reported scores (0.31 ~ 3.08) intended to be percentages?
> >
> > [1] DeepSeek-Prover-V2: Advancing Formal Mathematical Reasoning via Reinforcement Learning for Subgoal Decomposition

---

> > > ### Author Response · Authors · 2026-04-04
> > >
> > > We sincerely thank the reviewer for the prompt acknowledgement and for raising the overall score. We truly appreciate the thoroughness and constructiveness of the review, which has significantly improved our paper.
> > >
> > > Regarding the DeepSeek-ProverBench scores: thank you for catching this discrepancy. Our baseline data for DeepSeek-ProverBench were taken from the discussion section of Goedel-Prover-V2's OpenReview page [1]. To align with their evaluation setup, we only tested on the 15 AIME problems from the two most recent competitions within DeepSeek-ProverBench, which is also the setup adopted in Figure 1 of the original DeepSeek-Prover-V2 paper [2]. The scores in Table 7 of [2], however, are evaluated on the full ProverBench test set, which accounts for the numerical discrepancy. Since a full evaluation requires running all baseline models and our own models on the complete set, this effort is still underway. We will make every effort to include the full ProverBench results in the revision.
> > >
> > > [1] https://openreview.net/forum?id=j4C0nALrgK
> > >
> > > [2] DeepSeek-Prover-V2: Advancing Formal Mathematical Reasoning via Reinforcement Learning for Subgoal Decomposition

---

### Official Review · Reviewer_Bvpk · 2026-03-15

**Soundness:** 2
**Presentation:** 3
**Significance:** 3
**Originality:** 3
**Overall Recommendation:** 3
**Confidence:** 3

**Summary:**

The paper presents BFS-Prover-V2: a model trained to prove theorems in Lean in by predicting a proof step given a proof state. The model is then run with a best-first search to find complete proofs. The training is done with a typical expert iteration strategy, but a couple of interesting innovations are described: perplexity-based data curation and "soft resets." The inference is done via BFS, but it also involves planning agent to split the main goal into subgoals. The approach is evaluated on two established Lean benchmarks, miniF2F and ProofNet, and compared against multiple other proof-step and whole-proof generation models.

**Compliance With Llm Reviewing Policy:**

Affirmed.

**Key Questions For Authors:**

1. Can you characterize the difference in proving style between your BFS prover and Goedel-Prover-V2? Can you show examples of proofs for comparison?
2. Can you show some theorems from miniF2F that your prover solved but Goedel-Prover-V2 32B haven't?
3. When the planner provides a set of "have" statements, do you first check if these statements are actually sufficient for proving the main goal? Do you eliminate the redundant ones?
4. How exactly you run multiple agents for one have statement? Are these independent parallel runs?
5. Can you elaborate how cache works? Is it shared across all benchmark problems?
6. Why you didn't decide to use newer Lean version?
7. In Table 1, you report accumulative performance of the prover. What's the non-accumulative performance of the final checkpoint?
8. What's "scale up" in Fig. 4? If this is the 32B model, is it trained on data collected from 7B run?
9. What was the GPU/CPU setup for evaluation and how much time on average it took to evaluate for one benchmark problem?
10. What's the precise definition of a proof step in your experiments? For example, is `rw [x, y]` treated as one proofs step or two?
11. I'm not completely convinced of the effectiveness of the "soft reset" technique. Do you have a bit more data justifying it? Maybe continued training for a few more checkpoint would be better then restarting? Also, it would nice to see this on a plot.

**Limitations:**

yes

**Strengths And Weaknesses:**

**Strengths**

The paper concerns an exciting topic of mixing AI and formal theorem proving.

The authors focus on the proof-step generation approach, as opposed to a now more popular whole-proof generation approach. I find it a strength of the paper that this less popular paradigm is explored.

Some of the methods described by the authors are novel and inspiring.

**Weaknesses**
1. The main problem of the paper is this: given the presented results it's very unclear how the proof-step generating BFS-Prover-V2 compares with simpler whole-proof generation approaches like Goedel-Prover-V2. We would need to have some alignment of inference time budget for meaningful comparison in Table 1. Besides theoretical compute budget comparison it would be nice to have some estimates of practical wall-clock inference time. Proof-step models interact with Lean while generating, but this may cause huge and impractical slowdown.
2. The experiments could be more comprehensive. PutnamBench could be included. Godel-Prover-V2, as an opensource model, could be evaluated on ProofNet for comparison. Also, quite strangely, the author do not include BFS with Planner results for ProofNet.
3. Only accumulative performance is reported in Table 1, it would be good to also see non-cumulative performance of the final model.
4. The model was trained on quite old mathlib version (v4.10, released in Jul 2024). This means that some of the proof generated by the model won't compile in the newer version of mathlib.
5. Some details in the paper are missing: see questions.

---

> ### Author Rebuttal · Authors · 2026-03-31
>
> We thank the reviewer for the thorough review and address each concern below.
> ## Weaknesses
> 1. Each prover searches with a wall-time budget of 600 seconds, including all Lean interactions. Since Goedel-Prover-V2 does not report wall-time data, direct comparison is difficult, but per-problem time is on the same order. Although step-level provers require more frequent interactions, it may not be less efficient: (1) step-level provers tend to discover shorter proofs; (2) lower context memory enables 2-4x parallelism under identical compute. We will add a wall-time analysis on benchmarks in the revision.
> 2. We have conducted additional evaluations:
>
> |DeepSeek-ProverBench|Score|
> |---|---|
> |DeepSeek-Prover-V2-7B|0.31|
> |Kimina-Prover-Distill-8B|1.38|
> |Kimina-Prover-72B|1.53|
> |Goedel-Prover-V2-8B|1.53|
> |Goedel-Prover-V2-32B|1.85|
> |Goedel-Prover-V2-32B (w/ revision)|2.46|
> |BFS-Prover-V2-32B|1.85|
> |BFS-Prover-V2-32B (w/ planner)|3.08|
>
> |ProofNet|Score|
> |---|---|
> |DeepSeek-Prover-V2-7B|23.0|
> |Goedel-Prover-V2-32B|22.6|
> |Goedel-Prover-V2-32B (w/ RAG)|28.5|
> |DeepSeek-Prover-V2-671B|37.1|
> |BFS-Prover-V2-32B|41.4|
>
> (data partially from Goedel-Prover-V2's OpenReview comments.)
>
> PutnamBench contains errors on older Lean versions and requires substantial compute. We will make every effort to include PutnamBench in the camera-ready version.
>
> 3. BFS-Prover-V2-32B achieves 85.25% on miniF2F under 8192 × 3 × 600 and 40.32% on ProofNet under 8192 × 8 × 600, both via pure tree search. We will include the full benchmark-wise non-cumulative results in the revision.
>
> 4. This can be addressed by switching the Lean version used for interaction during tree search. Small-scale experiments show no significant performance degradation, partly because some tactics are strengthened in newer releases.
> 5. See below.
> ## Questions
> 1. Tree search tends to produce shorter, more tactic-efficient proof styles. With the planner, proofs exhibit clear hierarchical structure. Since Goedel-Prover-V2 does not include proof outputs in their report, we compare against DeepSeek-Prover-V2 in Appendix A. As Goedel-Prover-V2's cold-start data is distilled from DeepSeek-Prover-V2, these comparisons are representative of the broader stylistic difference.
> 2. Since Goedel-Prover-V2 has not published the list of miniF2F problems solved, direct problem-level comparison is infeasible. Examples that BFS-Prover-V2 solves via pure tree search but DeepSeek-Prover-V2 does not include: algebra_ineq_nto1onlt2m1on, amc12a_2020_p4, amc12a_2020_p7,etc. We will include a comprehensive comparison table in the revision.
> 3. We first run a syntax check to filter out invalid `have` statements. We do not verify sufficiency or remove redundancy; instead, if a subgoal is unprovable, we use dynamic replanning to produce an updated decomposition. This design keeps the architecture simple and robust to imperfect planning.
> 4. For each have statement, we launch multiple independent provers in parallel. When the first prover finds a proof, all others terminate and advance to the next subgoal.
> 5. The subgoal cache is per-problem. For each theorem it stores: subgoals from the planner; status of each (pending/proving/proven); tactics for solved subgoals. New prover instances replay solved subgoals to reach the correct state, then search from the first unsolved one. The cache is reset for each new problem.
> 6. Our pipeline uses Lean v4.10 with LeanDojo. Upgrading mid-project would require re-extracting all state-tactic pairs, re-running autoformalization, and re-validating proofs, a substantial engineering overhead. The core algorithmic contributions are version-independent and directly transferable to newer versions.
> 7. See weakness 3.
> 8. "Scale up" refers to the 7B to 32B transition at checkpoint 16. The 32B model is initialized from Qwen2.5-32B and trained on curated data from the 7B run. The 7B model approached capacity limits with diminishing returns. Scaling to 32B allowed us to (1) continue the iteration process with greater model capacity and (2) validate the scalability of our multi-stage approach. Detailed analysis is in Section 3.3.
> 9. We allocate 8 H20 GPUs with 128 CPUs, running 8 parallel 7B or 4 parallel 32B prover instances. Some problems are solved within a few interactions; others consume the full budget (pass@8192, 600s timeout per instance). Overall, the majority of problems fall on the shorter end.
> 10. A proof step = one model generation = one tactic as extracted by LeanDojo. E.g. rw [x, y] is one proof step.
> 11. Performance plateaus are visible at checkpoints 2, 6, 16, 19 in Figure 4. Continuing beyond leads to: (1) data scarcity, newly solved problems become extremely few; (2) diminishing/negative returns on downstream performance. Our soft reset addresses both. Section 3.3 shows quantitatively that retraining from the base checkpoint with accumulated high-quality data consistently breaks plateaus. We will expand with additional visualizations in the camera-ready version.

---

> > ### Author Rebuttal · Reviewer_Bvpk · 2026-04-04
> >
> > Thank you for the response, however I don't think it resolved all my concerns.
> >
> > 1. Goedel-Prover-V2 is open-source, so comparison with it could be more comprehensive, proof styles could be contrasted, wall-time could be compared, etc.
> > 2. Especially the lack of inference wall-time analysis is a big weakness of the work.
> > 3. Why no planner in ProofNet eval?
> > 4. More data on soft restarts missing.

---

> > > ### Author Response · Authors · 2026-04-05
> > >
> > > We sincerely thank the reviewer for the acknowledgement. We further clarify the remaining concerns and provide additional evidence.
> > >
> > > ## 1. Comparison with Goedel-Prover-V2
> > > As noted in our rebuttal, Goedel-Prover-V2's proof style is largely consistent with other whole-proof generation models, since its cold-start data is distilled from DeepSeek-Prover-V2. To provide the direct comparison the reviewer requested, we ran Goedel-Prover-V2-32B on the four problems whose BFS-Prover-V2 proofs are presented in Appendix A. The solutions are available in Goedel_V2_solutions folder at https://anonymous.4open.science/r/BFS-Prover-V2-365E, enabling a direct comparison between the two paradigms. Across all four problems, Goedel-Prover-V2 produces more verbose proofs with extensive `have` blocks, while BFS-Prover-V2's step-level tree search discovers significantly more concise proofs that leverage powerful tactics and theorems more effectively.
> > >
> > > ## 2. Wall-Time Analysis
> > >
> > > We would like to discuss why we believe it's not the most informative metric for comparing proving paradigms, while also clarifying what we can provide.
> > >
> > > - Wall-time is not a standardized comparison metric across models. Models of different sizes require fundamentally different compute resources, making wall-time comparisons conflated with hardware and parallelism choices rather than algorithmic efficiency. Moreover, for methods requiring many passes, reporting wall-time for only the successful pass is misleading, while reporting total wall-time across all passes is dominated by infrastructure-level factors (proportion of GPU/CPUs, degree of parallelism, per-tactic timeout) rather than by the proving method itself. We believe this is also why most open-source formal proving works, including InternLM-Prover, Hunyuan-Prover, MPS-Prover, DeepSeek-Prover series, Kimina-Prover series, and Goedel-Prover series do not report detailed wall-time analysis.
> > >
> > > - Step-level provers and whole-proof generation models serve partially different use cases. Beyond end-to-end proving, step-level provers can function as tactic-suggestion copilots, where the model suggests intermediate tactics during interactive proof development. This is not captured by end-to-end wall-time measurements. A demo of BFS-Prover-V2 used in this mode is available at https://anonymous.4open.science/r/BFS-Prover-V2-365E/demo.mp4.
> > >
> > > That said, to provide some concrete data points, we measure the wall-time of Goedel-Prover-V2-32B on the four problems tested in 1. The average wall-time per pass was approximately 917 seconds, while BFS-Prover-V2's evaluation uses a 600-second timeout per prover instance (which includes all Lean interaction time). We will include more comprehensive wall-time statistics in the revision. We also want to emphasize that BFS-Prover-V2's lower per-step context length naturally enables 2-4x higher parallelism and faster inference rate, offsetting the overhead from more frequent Lean interactions.
> > >
> > > ## 3. Planner on ProofNet
> > >
> > >  We acknowledge that including planner results on ProofNet would further strengthen the evaluation. The primary purpose of the ProofNet experiments is to demonstrate the strong OOD generalization of our model. Even without the planner, BFS-Prover-V2 achieves 41.4% on ProofNet via pure tree search, surpassing all reported open-source results. This already provides strong evidence for the OOD generalization we aim to demonstrate.
> > >
> > > The planner experiments on ProofNet are currently underway and we will add these in the revision.
> > >
> > > ## 4. Soft Restart Effectiveness
> > >
> > > We respectfully note that the existing evidence for soft restarts is already substantial. In Figure 4, soft restarts are applied at multiple points throughout the training process, and in every case the pattern is consistent in that performance plateaus before the restart and improves afterward. This pattern is replicated across:
> > >
> > > - Training stages: early (ckpt 3), mid (ckpt7), and late (ckpt 17,20)
> > > - Model sizes: 7B (ckpt 3,7) and 32B (ckpt 17,20)
> > > - Quantitative measurements: Section 3.3 provides numerical analysis showing each soft restart breaks the preceding plateau
> > >
> > > These independent plateau-recovery events at different scales provide strong empirical evidence. Running a fully controlled ablation (a parallel training run without soft restarts) requires replicating the 22-round expert iteration pipeline, which would be disproportionately expensive relative to the evidence it adds beyond the multi-condition consistency already demonstrated.
> > >
> > > We hope this additional clarification addresses the reviewer's remaining concerns. We believe the current submission already provides strong evidence that BFS-Prover-V2 advances the the state of the art for step-provers through novel training and inference techniques, and we are committed to open-sourcing the system and adding additional results discussed above in the revision. We would be grateful if the reviewer could consider raising the score accordingly.

---

### Decision · Program_Chairs · 2026-04-30

**Decision:**

Accept (regular)

**Comment:**

Two reviewers recommend Accept (5) and one recommends Weak Reject (3). The average is 4.33, and I recommend Accept.

BFS-Prover-V2 introduces two innovations for LLM-based step-level theorem proving: a multi-turn off-policy RL framework with perplexity-based tactic filtering and soft resets to overcome training plateaus, and a planner-enhanced multi-agent system for inference-time scaling via subgoal decomposition and shared caching. The results are strong — 95.08% on miniF2F and 41.4% on ProofNet, with the ProofNet result surpassing all reported open-source baselines including much larger models. The paper makes a meaningful case for reviving proof-step generation as a competitive paradigm alongside whole-proof generation.

The two accepting reviewers (gnrk and 7rcS) were satisfied after the rebuttal, which clarified the definition of "accumulative" performance, provided budget-aligned comparisons with Goedel-Prover-V2 on DeepSeek-ProverBench and ProofNet, addressed data contamination concerns, clarified the human data origin (state-tactic pairs from Mathlib), and committed to open-sourcing the model weights and inference code.

The weak rejecting reviewer (Bvpk) maintained their score, raising concerns about: (1) no wall-clock inference time comparison with Goedel-Prover-V2; (2) no planner results on ProofNet; (3) limited ablation evidence for soft resets. These are legitimate presentation and evaluation gaps that the authors should address in the camera-ready version — the authors have committed to wall-time analysis and ProofNet planner results. The soft restart evidence, while not a fully controlled ablation, is presented consistently across multiple plateau-recovery events at different model scales, which is reasonable given the cost of rerunning a 22-round expert iteration pipeline.

On balance, the technical contributions are solid and well-validated, the benchmark results are state-of-the-art for step-level provers, and the rebuttal addressed the most substantive concerns. Accept.